# PowerGraph: A power grid benchmark dataset for graph neural networks

**Anna Varbella**[*]
D-MAVT
ETHZ
Zurich, Switzerland
avarbella@ethz.ch

**Kenza Amara**[*]
D-INFK
ETHZ
Zurich, Switzerland
kenza.amara@ai.ethz.ch

**Blazhe Gjorgiev**
D-MAVT
ETHZ
Zurich, Switzerland
gblazhe@ethz.ch

**Mennatallah El-Assady**
D-INFK
ETHZ
Zurich, Switzerland
menna.elassady@ai.ethz.ch

**Giovanni Sansavini**
D-MAVT
ETHZ
Zurich, Switzerland
sansavig@ethz.ch

## Abstract

Power grids are critical infrastructures of paramount importance to modern society and, therefore, engineered to operate under diverse conditions and failures. The ongoing energy transition poses new challenges for the decision-makers and system operators. Therefore, developing grid analysis algorithms is important for supporting reliable operations. These key tools include power flow analysis and system security analysis, both needed for effective operational and strategic planning. The literature review shows a growing trend of machine learning (ML) models that perform these analyses effectively. In particular, Graph Neural Networks (GNNs) stand out in such applications because of the graph-based structure of power grids. However, there is a lack of publicly available graph datasets for training and benchmarking ML models in electrical power grid applications. First, we present PowerGraph, which comprises GNN-tailored datasets for i) power flows, ii) optimal power flows, and iii) cascading failure analyses of power grids. Second, we provide ground-truth explanations for the cascading failure analysis. Finally, we perform a complete benchmarking of GNN methods for node-level and graph-level tasks and explainability. Overall, PowerGraph is a multifaceted GNN dataset for diverse tasks that includes power flow and fault scenarios with real-world explanations, providing a valuable resource for developing improved GNN models for node-level, graph-level tasks and explainability methods in power system modeling. The dataset is available at https://figshare.com/articles/dataset/PowerGraph/22820534 and the code at https://github.com/PowerGraph-Datasets

## 1  Introduction

Our modern society depends on a reliable power system [1], essential for daily life and economic activities. Power systems are engineered to operate under diverse conditions and failures. However, the increasing complexity of power systems, driven by electrification and the rise of intermittent energy sources, is posing new challenges. Transmission system operators (TSOs) require online tools for effective power systems monitoring, but the current methods for grid analyses, hindered by their computational speed, cannot fully meet this need. Additionally, the failure of critical

---

[*]Equal Contribution

38th Conference on Neural Information Processing Systems (NeurIPS 2024) Track on Datasets and Benchmarks.

components under specific conditions can trigger cascading outages, potentially leading to complete blackouts of the power grid [2, 3]. Due to the rarity of such events and the scarcity of historical data, computer models are used to simulate cascading failures. These models replicate the complex behavior of systemic responses and the propagation of successive failures within the grid. However, the computational intensity of these tools prevents their use by power grid operators for real-time detection of cascading failures.

Machine learning techniques, particularly GNNs, offer significant potential for providing real-time solutions in electrical power systems and are well-suited for modeling power grid phenomena [4]. Works such as [5, 6] address the power flow (PF) problem by employing machine learning to replace traditional solvers. Similarly, researchers in [7, 8] explore replacing non-linear solvers for optimal power flow (OPF) problems. These endeavors often test methods on synthetic power systems or benchmark IEEE power grids. In the realm of fault scenario applications, the need for a real-time tool capable of estimating the potential of cascading failures under various power grid operating conditions is evident. Although recent methodologies employing machine learning algorithms for real-time prediction of cascading failures show promise, they often lack generalizability across diverse failure sets [9, 10] or use less accurate linear approximations of the AC power flow[11]. Addressing these limitations, [12] demonstrates the superiority of GNNs over FNNs for power grids, albeit without providing a complete dataset or model benchmarking across diverse power grids.

The lack of explainability often hinders the application of ML tools in specific industries such as the power systems sector. While synthetic graph data generators have made strides in providing benchmark datasets with ground-truth explanations [13], none offer real-world data with empirical explanations in the context of graph classification. Power systems practitioners could greatly benefit from explanations of black-box deep learning models. Firstly, interpretable models are inherently more trustworthy. Secondly, if GNN explainability models could identify the specific edges causing a cascading failure, TSOs would gain critical insights into the power system's most vulnerable components.

Publicly available power grid datasets, such as the Electricity Grid Simulated (EGS) datasets [14], the PSML [15], and the Simbench dataset [16], are not tailored for machine learning on graphs. Some efforts, like [17], focus on the dynamic stability of synthetic power grids. Furthermore, we identified a critical gap in accordance with the OGB taxonomy for graph datasets [18, 19], where no GNN dataset in the society domain is available for graph-level tasks.

In this work, we present PowerGraph, aiming to address the following research gaps:

- A public dataset designed for various power systems to solve power flow and optimal power flow problems using GNN supervised learning techniques.
- A public dataset encompassing a wide range of cascading failure scenarios for different graph-level tasks.
- A real-world dataset for GNN graph-level tasks with clear ground-truth explanations for GNN explainability.

The PowerGraph dataset comprises GNN-tailored datasets for i) power flows, ii) optimal power flows, and iii) cascading failure analyses of power grids. To create this comprehensive dataset, we leverage MATPOWER [20] for power flow and optimal power flow simulations and Cascades [21], an alternate-current (AC) physics-based model for cascading failure analyses. This process ensures that our dataset encompasses many scenarios, enabling robust analysis and training of GNN models for power grid applications. As a result, PowerGraph is a large-scale graph dataset tailored for power flow analysis and the prediction of cascading failures in electric power grids, modeled as classification or regression problems at the node and graph level. The breakdown of the total number of graphs across node and graph-level tasks per power grid is detailed in Table 1.

The PowerGraph dataset encompasses the IEEE24 [23], IEEE39 [24], IEEE118 [25], and UK transmission system [26]. These selected test power systems mirror real-world-based power grids, offering a diverse array of scales, topologies, and operational characteristics. They provide comprehensive data essential for conducting cascading failure analysis. With PowerGraph, we aim to democratize the use of Graph Neural Networks (GNNs) in critical infrastructures like power grids. Our contributions are as follows:

- Introducing a data-driven approach for analyzing power flow and cascading failure events in power grids in real-time.

Table 1: The PowerGraph dataset serves both node and graph-level tasks. Electrical buses symbolize nodes, while transmission lines and transformers represent edges. We detail parameters for power flow and cascading failure analyses on four chosen power grids. The loading condition data [22], for a period of one year, are provided at a 15-minute resolution, with each graph illustrating a single loading condition. The cascading failure analysis data is given as a set of graphs, each of which denotes the power grid loading condition linked to a triggering outage.

| Test system | No. Bus | No. Branch | Power flow analysis-Node level tasks | Cascading failure analysis-Graph level tasks |
|---|---|---|---|---|
| | | | No. Graphs-N | No. Graphs- N |
| IEEE24 | 24 | 38 | 34944 | 21500 |
| UK | 29 | 99 | 34944 | 64000 |
| IEEE39 | 39 | 46 | 34944 | 28000 |
| IEEE118 | 118 | 186 | 34944 | 122500 |

- Assessing and benchmarking various GNN architectures and hyperparameters.
- Providing the first real-world GNN dataset with empirical explanations to benchmark GNN explainability methods.
- Making the dataset easily accessible in a user-friendly format, allowing the GNN community to experiment with different architectures for node and graph-level applications.
- Offering a range of tasks at both the node and graph levels, including regression, binary classification, and multi-class classification.

The paper is structured as follows: Section 2 outlines the dataset structure; Section 3 presents the benchmark of GNN for power flow analysis, optimal power flow and cascading failure analysis, Section 4 presents explainability methods for cascading failure analysis; and Section 6 concludes with final remarks and discussion.

## 2 The PowerGraph dataset

### 2.1 Node-level task: power flow and optimal power flow

The PowerGraph dataset for node-level tasks is obtained by performing power flow (PF) and optimal power flow (OPF) simulations with MATPOWER [20], using the formulations in Sections A.2.1 and A.2.2. This dataset consists of $N$ attributed graphs, denoted as $\mathcal{G} = G_1, G_2, \ldots, G_N$, with each graph representing a different grid loading condition. An attributed graph is defined as $G = (\mathcal{V}, \mathcal{E}, \mathbf{V}, \mathbf{E})$, where $\mathcal{V}$ is the set of nodes (buses, i.e., the power injection points in the grid), and $\mathcal{E}$ is the set of edges (branches, i.e., the power transmission asset in the grid). The node feature matrix $\mathbf{V} \in \mathbb{R}^{|\mathcal{V}| \times t}$ includes $t$ features for each of the $|\mathcal{V}|$ nodes, and the edge feature matrix $\mathbf{E} \in \mathbb{R}^{|\mathcal{E}| \times s}$ includes $s$ features for each of the $|\mathcal{E}|$ edges. Figure 1 illustrates the dataset structure for both PF and OPF tasks. PF analysis aims to solve the power flow equations, given the power dispatched at each generator except for a balancing bus called the slack bus. The predicted node-level features depend on the type of electrical bus, as detailed in Section A.2.1. The OPF problem aims to determine the optimal generator (dispatch) to operate the grid at minimum cost while respecting physical constraints. Here, too, the predicted node-level features depend on the type of bus, as detailed in Section A.2.2.

### 2.2 Graph-level task: cascading failure analysis

The PowerGraph dataset for graph-level tasks is obtained by processing the results of the Cascades model (details in Section A.3). This dataset consists of $N$ attributed graphs, each representing a unique pre-outage operating condition and a set of outages involving single or multiple branches. The total number of graphs $N$ per power grid equals $n_{load\ cond} * n_{outlists}$, with specific values listed in Table 8. Outages result in the removal of corresponding branches from the adjacency matrix and edge features, altering the graph topology. Figure 2 shows an example of the PowerGraph dataset for a power grid. This structure is consistent across all grids in PowerGraph, including IEEE24, IEEE39, UK, and IEEE118. The total number of instances is in Table 1. The post-outage evolution is known for each initial state, including demand not served (DNS) and the number of tripped lines, i.e. the edges that have failed during the cascading failure (henceforth called cascading edges). A cascading failure occurs if any additional branch trips after the initial outage.

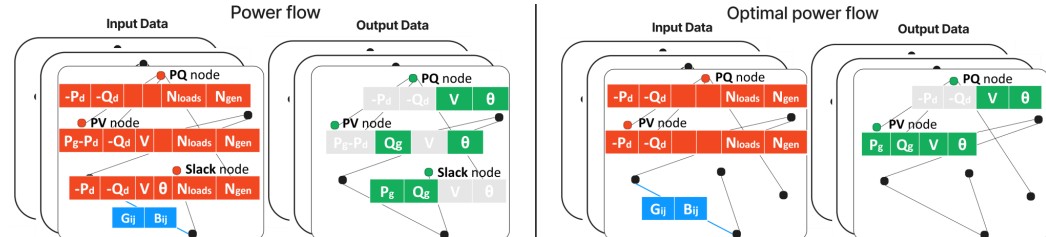

Figure 1: Instance of the PowerGraph dataset for power flow and optimal power flow. The input node features are in red, and output node-level predictions are in green. The known input quantities are reported for different node types, and the unknown quantities are set to zero to maintain the dataset's dimensionality structure (indicated by an empty cell in the picture). Similarly, the output quantities depend on the node type; if a variable is known, we mask it during training, and masked values are indicated with grey cells. The quantities are: active power generation $P_g$, reactive power generation $Q_g$, active power demand $P_d$, reactive power demand $Q_d$, voltage magnitude $V$, and voltage angle $\theta$, the number of loads $N_{loads}$, and number of generators $N_{gen}$. The edge level features are: branch conductance $G_{ij}$ and branch susceptance $B_{ij}$.

Each graph is assigned an output label depending on the task. For **binary classification**, the graphs are labeled as stable (DNS=0) or unstable (DNS>0). In **multi-class classification**, the graphs are categorized into four classes shown in Table 2. For **regression** tasks, the label corresponds to the DNS in MW. The choice among binary classification, multi-class classification, or regression depends on the use case of the GNN model trained with the PowerGraph dataset. Binary classification models serve as early warning systems to detect critical grid states. Multi-class models distinguish different scenarios, helping transmission system operators understand when a cascading failure might not lead to unserved demand. Regression models estimate the DNS for specific pre-outage states, acting as surrogates for physics-based models and enabling security evaluations with low computational costs.

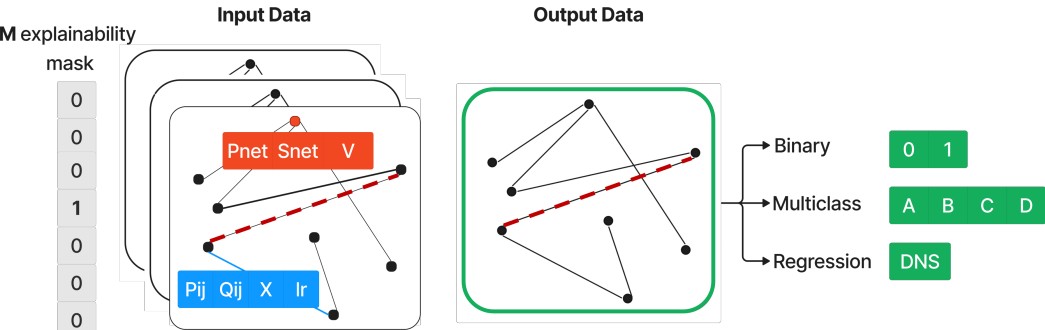

Figure 2: Instance of the PowerGraph dataset for cascading failure analysis. We highlight the initial outage with the red-dotted line, which is removed from the graph connectivity matrix and from the edge feature matrix. The cascading edge is in bold and encoded in the **M** boolean vector (0 - the edge has not tripped during cascading development, 1 - otherwise). The input node features are the: net active power $P_{net} = P_{gen} - P_{load}$, net apparent power $S_{net} = S_{gen} - S_{load}$, and voltage magnitude $V_i$. Where $P_{gen}$ and $P_{load}$ are the active power generation and demand, respectively, and $S_{gen}$ and $S_{load}$ are the apparent power generation and demand, respectively. The input edge features are: active power flow $P_{i,j}$, reactive power flow $Q_{i,j}$, line reactance $X_{i,j}$, and line rating $lr_{i,j}$.

Table 2: Multi-class classification of datasets. c.f. stands for *cascading failure* and describes a state resulting in cascading failure of components. DNS denotes demand not served.

| Category A | Category B | Category C | Category D |
|---|---|---|---|
| DNS > 0 MW | DNS > 0 MW | DNS = 0 MW | DNS = 0 MW |
| c.f. occurs | no c.f. | c.f. occurs | no c.f. |

**Explainability mask** We assign ground-truth explanations as follows: when a system state undergoes a cascading failure, the cascading edges are considered to be explanations for the observed demand not served. Therefore, for the Category A instances, we record the branches that fail dur-

Table 3: Results of categorization in percentage.

| Power grid | Category A | Category B | Category C | Category D |
|---|---|---|---|---|
| IEEE24 | 15.8% | 4.3% | 0.1% | 79.7% |
| IEEE39 | 0.55% | 8.4% | 0.45% | 90.6% |
| UK | 3.5% | 0 | 3.8% | 92.7% |
| IEEE118 | >0.1% | 5.0% | 0.9% | 93.9% |

ing the development of the cascading event. We set the explainability mask as a Boolean vector $\mathbf{M} \in \mathbb{R}^{|\mathcal{E}| \times 1}$, whose elements are equal to 1 for the edges belonging to the cascading stage and 0, otherwise (see Figure 2).

## 3    Benchmarking the PowerGraph dataset

In this section, we present the experimental setting to benchmark the PowerGraph dataset for node-level tasks (i.e. power flow and optimal power flow analysis) and graph level tasks (i.e. cascading failure analysis).

**Experimental Setting and Evaluation Metrics**    For each power grid dataset, both for graph and node-level tasks we utilize four baseline GNN architectures: GCNConv [27], GATConv [28], GINEConv [29], and TransformerConv [30], which are commonly used in the graph xAI community. We experiment with state-of-the-art graph transformer convolutional layers [30], forming the backbone of recent models such as GraphGPS [31], Transformer-M [32], and TokenGT [33], chosen for their ability to incorporate edge features relevant to power grids. The GNN architecture includes multiple message passing layers (MPLs), each followed by a PReLU activation function [34]. When graph-level tasks are involved, a maximum global pooling operator for obtaining graph-level embeddings from node embeddings, and one fully connected layer. We perform a grid search over the number of MPLs (1, 2, 3) and the hidden dimensionality (8, 16, 32). The Adam optimizer is used with an initial learning rate of $10^{-3}$, and each model is trained for 50 epochs, with the learning rate adjusted using a scheduler that automatically reduces it if the reference metric stops improving. The datasets are split into 85% train, 5% validation, and 10% test sets, with a batch size of 32 for node-level tasks and 16 for graph-level tasks. For classification models, balanced accuracy [35] is used due to class imbalance, (see Table 3). For regression models, mean squared error is used as the reference metric.

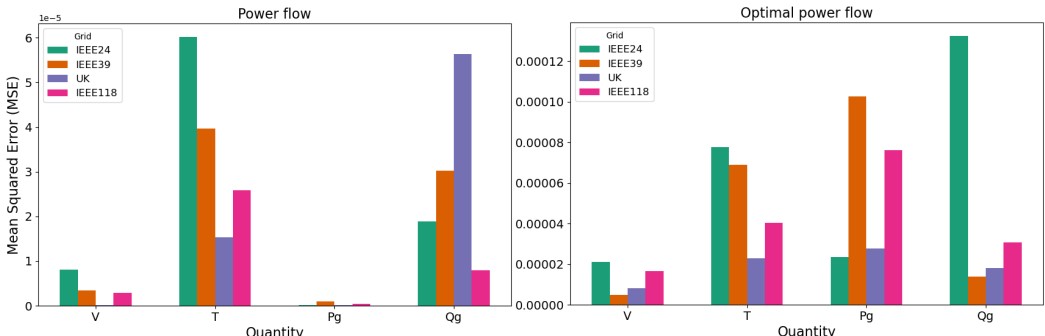

Figure 3: Node-averaged Mean Absolute Errors on the predicted physical quantities for the power flow and optimal power flow problems on the best performing models reported in Table 4.

**Observations**    Table 4 presents the best models for node-level tasks using GCN, GAT, GINe, and Transformer architectures across the four power systems. Similarly, Table 5 reports the top-performing models for graph-level tasks. In addition to the presented result in this paper and its appendices, we present the complete results, including all models and their performances, on our website: `https://powergraph.ivia.ch`. The website also includes Gradient Boosted Trees results as a baseline for comparison. Figure 3 illustrates the node-averaged MSE for each predicted physical quantity for the power flow tasks for the best performing models reported in Table 4.

**Discussion**  We train our models using data from current simulation methods (see A.2.1 and A.3), aiming for the highest accuracy in classification and regression to replicate those results, while the key advantage of our ML models lies in surpassing traditional solvers or simulations in computational efficiency. For node-level tasks, the OPF task presents a higher level of complexity, leading to a higher mean squared error (MSE) for the machine learning model trained on it compared to the model trained on the PF task, as shown in Figure 1 and Table 4. Although model performance varies slightly across different grids, we observe that GNNs are effective in predicting optimal power flow and power flow solutions across grids with varying topologies, achieving low prediction error for key physical quantities regardless of grid size.

For both graph-level and node-level tasks, the Transformer model consistently outperforms other approaches, achieving optimal results, particularly with three MPL layers in most cases. In contrast, the GCN model repeatedly demonstrates lower performance compared to other models, emphasizing its limitations for power flow tasks where edge features are critical for analyzing power flow and modeling potential cascades. This highlights the advantages of the Transformer and GINe models, which emerge as the top performers. The Transformer's attention mechanism allows it to dynamically assign importance to neighboring nodes, capturing intricate relationships that enhance its predictive accuracy. This attention-based approach significantly contributes to the effectiveness of both the Transformer and GAT models across node-level and graph-level tasks. Meanwhile, the GINe model also shows strong performance due to its powerful ability to capture the graph topology of power grids, further boosting its predictive accuracy [29].

While binary and multi-class classification models show good results, the regression model predicting the exact demand not served does not. For graph-level regression tasks, the R2 [36] score, also known as the coefficient of determination, peaks at 0.43 for the best model on the IEEE24 dataset but falls below 0.26 across other power systems. Ideally, an R2 score closer to 1 is desired for a better model fit, underscoring the need for further research to enhance GNN performance on regression tasks. on improving GNN performance on regression tasks. Additionally, GNNs show limitations on node-level regression, where simpler models like Gradient Boosted Trees (GBT) sometimes outperform them on certain datasets (see `https://powergraph.ivia.ch`). However, this trend does not extend to graph-level tasks, where GNNs consistently prove to be the superior method. This calls for the development of GNN architectures specifically optimized for regression tasks, crucial for power systems analysis. At the same time, even minor improvements in classification tasks should not be underestimated, as they can significantly impact decision-making in critical infrastructures such as power grids.

Table 4: Power flow and optimal power flow model results on the test set, averaged over five random seeds. MSE error is used as a reference metric, and only the best model architecture results are reported.

| Power system | Task | Best model | MSE |
|---|---|---|---|
| IEEE24 | *Powerflow* | transformer 3h 32n | 4.35e-05±7.46e-06 |
| | *Optimal powerflow* | transformer 3h 32n | 8.99e-05±3.43e-06 |
| IEEE39 | *Powerflow* | transformer 3h 32n | 4.35e-05±1.27e-05 |
| | *Optimal powerflow* | transformer 3h 32n | 7.38e-05±7.87e-06 |
| IEEE118 | *Powerflow* | transformer 3h 32n | 2.39e-05±3.74e-06 |
| | *Optimal powerflow* | transformer 3h 32n | 5.74e-05±4.24e-06 |
| UK | *Powerflow* | transformer 3h 32n | 2.02e-05±8.68e-06 |
| | *Optimal powerflow* | transformer 3h 32n | 3.75e-05±1.39e-05 |

## 4  Benchmarking explanations on the graph-classification models

**Experimental setting and evaluation metrics**  The PowerGraph benchmark with explanations tests and compares explainability methods. Explanations are subsets of the grids. We prefer generating graph-based explanations over textual ones, as graph structures are more intuitive and effective for conveying critical nodes and edges to experts in transmission grids. The role of explainers is to identify the edges that are necessary for the graphs to be classified as Category A. This choice is explained in Appendix A.4. Then, the resulting edges are evaluated on how well they match the explanation masks, which represent the cascading edges, using the accuracy score, and on how they contribute to the model's prediction, using the faithfulness metric. The accuracy score refers to the balanced accuracy metric, defined in Appendix A.6, between the ground truth target cascading failure edges and the estimated explanatory edges by the xAI method. The faithfulness score measures the changes

Table 5: Cascading failure analysis results on the test set averaged over five random seeds. The reference metric for the classification tasks is balanced accuracy, and the regression task is MSE. Only the best model architecture results are reported

| Power system | Task | Best model | Metric | |
|---|---|---|---|---|
| **IEEE24** | *binary* | transformer 3h 32n | Balanced accuracy | 0.9828±0.0056 |
| | *multiclass* | transformer 3h 32n | Balanced accuracy | 0.9828±0.0056 |
| | *regression* | gin 3h 32n | MSE | 7.82e-04±1.62e-04 |
| **IEEE39** | *binary* | transformer 2h 32n | Balanced accuracy | 0.9880±0.0020 |
| | *multiclass* | transformer 3h 32n | Balanced accuracy | 0.9765±0.0064 |
| | *regression* | transformer 2h 16n | MSE | 6.31e-05±1.86e-05 |
| **IEEE118** | *binary* | gin 3h 32n | Balanced accuracy | 0.9982±0.0006 |
| | *multiclass* | gin 3h 32n | Balanced accuracy | 0.9962±0.0014 |
| | *regression* | gin 3h 32n | MSE | 2.99e-06±3.19e-06 |
| **UK** | *binary* | gin 3h 32n | Balanced accuracy | 0.9951±0.0022 |
| | *multiclass* | gin 2h 32n | Balanced accuracy | 0.9845±0.0019 |
| | *regression* | transformer 3h 32n | MSE | 1.07e-03±3.47e-04 |

in model outputs induced by retraining only with the important graph entities identified, and its definition is given in Appendix B.5. We compare the results obtained for each of the four PowerGraph datasets: IEEE24, IEEE39, IEEE118, and UK. For each dataset, we select the trained Transformer model with 3 layers and 16 hidden units, obtained with the configuration of Section 3. To benchmark explainability methods, having the best GNN model is not essential. By appropriately filtering predictions (correct or mixed) and focusing the explanations (on the phenomenon or model) [37], we can work around lower test accuracy. We compare non-generative methods, including the heuristic Occlusion [38], gradient-based methods Saliency [39], Integrated Gradient [40], and Grad-CAM [41], and perturbation-based methods GNNExplainer [42], PGMExplainer [43] and SubgraphX [44]. We also consider generative methods: GSAT [45], D4Explainer [46] and RCExplainer [47]. For more details on generative vs. non-generative explainers, see Appendix B.2 and B.3. We compare those explainability methods to the base estimators: Random, Truth, and Inverse. Random assigns random importance to edges following a uniform distribution. Truth estimates edge importance as the pre-defined ground-truth explanations of the datasets. i.e., the cascading edges. The Inverse estimator represents the worst-case scenario, assigning edges opposite to the ground truth. Appendix B provides additional experimental details on Transformer performance and the explainability methods.

Table 6: Explainability results summary, using the number of explained instances and the number of ground-truth edges, i.e., the cascading failure edges.

| Power System | IEEE24 | UK | IEEE39 | IEEE118 |
|---|---|---|---|---|
| **No. Explained grids** | 3416 | 2236 | 154 | 11 |
| **Avg No. cascading edges** | 2.4 | 6.8 | 3.1 | 3.4 |
| **Max No. cascading edges** | 8 | 10 | 10 | 6 |
| **Ratio GT edges/Total edges** | 0.2 | 0.1 | 0.2 | 0.03 |

**Human-centric evaluation** We use the balanced accuracy metric to evaluate the generated explanations. It compares the generated edge mask to the ground-truth cascading edges and takes into account the class imbalance since the cascading edges are a small fraction of the total edges. It measures how convincing the explanations are to humans. Appendix A.6 gives more details about this metric. We report the performance of 14 explainability methods on finding ground-truth explanations. All results are averaged on five random seeds.

Figure 4 shows the results of the human-centric evaluation. The Truth explainer consistently achieves an accuracy score of 1, while the Inverse method performs significantly worse with an accuracy score of 0.5, as expected. Additionally, the Random method exhibits similarly low accuracy scores, which can be attributed to the small amount of cascading edges within the grids (see Table 6). Regarding the IEEE39 and IEEE118 grids, none of the explainability methods succeed in identifying the cascading edges within their $topk^*$ explanatory edges. The low performance of xAI methods on balanced accuracy for IEEE39 and IEEE118 is likely due to Category A graphs representing less than 1% of the

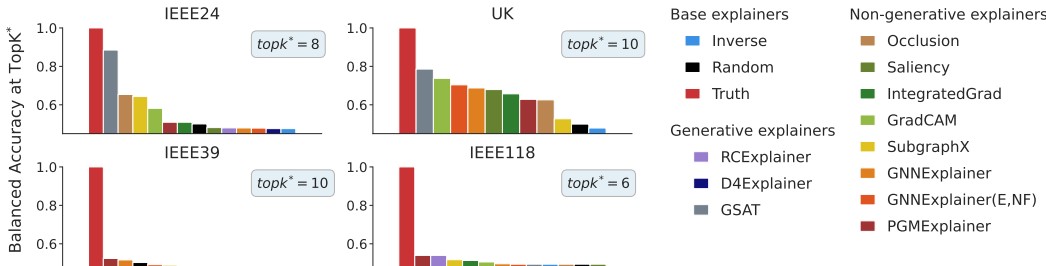

Figure 4: Balanced accuracy of the explanations with $topk^*$ edges. The *top* balanced accuracy is computed on explanatory edge masks that contain the $topk^*$ edges that contribute the most to the model predictions, with $topk^*$ being the number of edges in the corresponding ground-truth explanations, i.e. the maximum number of cascading edges for each dataset.

datasets, see Table 3. Resulting in a low number of explainable instances, i.e., 154 and 11 explainable graph instances, respectively. For the IEEE24 and UK datasets, which have more instances of Category A, GSAT significantly outperforms other methods, achieving a balanced accuracy of 0.9 for IEEE24 and almost 0.8 for UK. GradCAM and Occlusion methods are also able to identify a few cascading edges in these cases.

**Model-centric evaluation** Human evaluation is not always practical because it requires ground truth explanations and is subjective to human judgement, not necessarily accounting for the model's reasoning. Model-focus evaluation, however, measures the consistency of model predictions, removing or keeping the explanatory graph entities. We, therefore, evaluate the faithfulness of the explanations using the fidelity+ metric. The fidelity+ measures how necessary the explanatory edges are to the GNN predictions. For PowerGraph, edges with high fidelity+ are the ones necessary for the graph to belong to Category A. The $fid_+^{prob}$ is computed as in[37] and described in Appendix B.5. We follow the systematic evaluation framework GraphFramEx [37], more details found in B.4.

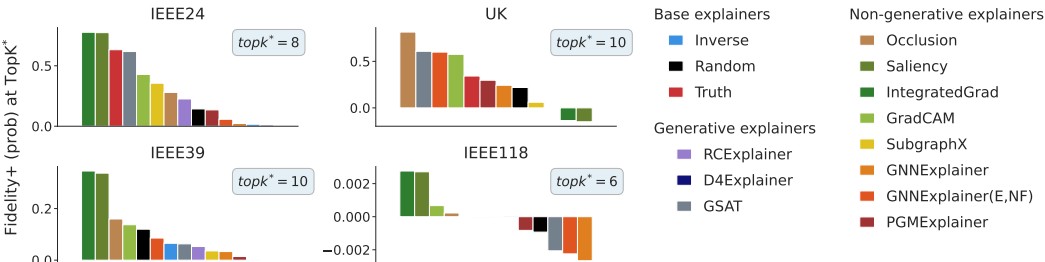

Figure 5: Faithfulness of the explanations with $topk^*$ edges. The faithfulness score is measured with the $fid+^{acc}$ metric as defined in Equation 7 in Appendix B.5. The optimal number $topk^*$ of edges kept for the explanations corresponds to the maximum number of expected cascading edges (i.e., ground truth explanations) and depends on the dataset.

Figure 5 shows the results of the model-centric evaluation of the explainability methods. Gradient-based methods generate the most faithful explanations between the Powergraph datasets. For the three IEEE datasets, IntegratedGrad and Saliency are the most faithful explainers; for the UK dataset, Occlusion and GradCAM give faithful results. These observations are consistent with the conclusions in [37], i.e. gradient-based xAI methods return more faithful explanations. Nevertheless, we observe low faithfulness scores for IEEE39 and IEEE118. Particularly for IEEE118, all methods show an average faithfulness score close to zero. Again, this might originate from the low ratio of cascading edges compared to the total number of edges in IEEE118 (refer to Table 6 for details). In addition, IEEE118 is the largest power grid with 186 branches, containing complex interdependencies among its power grid elements during cascading failures. Consequently, node and edge-level features play a significant role in explaining the GNN predictions. Therefore, we believe that an accurate model explanation will be obtained only with methods that provide node and link-level feature masks, along with edge masks.

**Discussion**    When comparing the outcomes of the human-centric evaluation in Figure 4 and the model-centric evaluation in Figure 5, a trend emerges: most methods exhibit contrasting performances when evaluated on the human-centric accuracy or the model-centric faithfulness. This means that explainability methods that excel in accuracy may not necessarily produce faithful explanations, and vice versa. In the model's perspective, cascading edges are not always necessary for predicting Category A events, i.e., DNS>0 and cascading failures. The ground-truth cascading edges (represented by the Truth baseline) reveal that they are not faithful to the model for both IEEE39 and IEEE118, and they are only marginally faithful for the UK dataset. Furthermore, explanations generated by faithful methods like Saliency and IntegratedGrad do not include any cascading edges. These observations highlight significant differences between human explanations based on physics-based simulations and model-centric explanations. Only GSAT generates accurate and faithful explanations, bridging the gap between human and model perspectives.

## 5    Limitations

While this study provides a valuable benchmark for power grid analysis using GNN models, there are limitations that should be acknowledged. Using power grid models, such as the IEEE test case systems, is a standard practice in power engineering modeling and analyses. Although these power systems are not ideal, they are still based on real power grids. The main reason we utilize these grids in our study is because real power system data is safety-critical and often classified, making it inaccessible for research. Furthermore, cascading failures are rare events, and data on their progression and the conditions under which they occur are largely unavailable. As a result, synthetic data is the only viable option for cascading failure analysis, which could limit the generalizability of the findings to real-world scenarios. Nevertheless, we use the Cascades model to simulate cascading failures and generate GNN datasets, which have been validated with historical blackout data from the WECC grid [48]. Furthermore, the small number of buses in the IEEE test cases may constrain the ability of the models to capture long-range dependencies, potentially affecting their performance on larger, more complex power grids. While this work demonstrates a method for modeling power systems with GNNs, other approaches could be explored in future research to enhance model accuracy and robustness. For instance, incorporating energy price variations to analyze the multi-period economic dispatch, multi-period OPF and unit commitment could provide a more comprehensive understanding of the economic aspects of power grid operations, which was not considered in this study

## 6    Conclusions

PowerGraph addresses critical challenges encountered in utilizing Graph Neural Networks (GNNs) for power systems analyses. PowerGraph effectively fills a significant gap in the GNN domain for power engineering applications by offering a dataset tailored for node and graph-level tasks and model explainability. Through rigorous benchmarking against a range of GNN and explainability models, PowerGraph exhibits high performance in graph classification, albeit indicating a need for further refinement in regression models. Regression models remain essential in power systems and can be solved by GNNs for tasks like power flow and system security analysis. However, GNN still encounters challenges that demand further research and development. Across power grid analysis tasks, we find that an architecture with three message-passing layers yields the most accurate predictions for both power flow and cascading failure analysis. This result suggests that incorporating information from up to three-hop neighbors is essential for precise modeling, effectively balancing local and global network information required for accurate predictions in both graph- and node-level tasks. Furthermore, PowerGraph is the first real-world dataset to offer ground-truth explanations for graph-level tasks in explainable AI, enabling thorough evaluation of various explainability methods. Despite this advancement, the performance of these methods remains suboptimal, highlighting the lack of a universally superior approach. Given the crucial role of explainability for power grid operators, this underscores the ongoing need for dedicated research and development in this field. Looking forward, we plan to enhance PowerGraph by adding a temporal graph dataset to facilitate in-depth analysis of the stages of cascading failures. Although this information is already present in the dataset, please refer to A.5, it was beyond the scope of this work. We also benchmark methods on larger synthetic power systems, such as the grid in [49], to assess whether deeper GNN architectures are necessary, as noted in [50]. A raw dataset and the benchmarking results are available at https://powergraph.ivia.ch.

## Acknowledgments

I would like to express my heartfelt gratitude to Rachel Wolfisberg, Louisa Hillegaart, Annamalai Lakshmanan, and Matteo Mazzonelli for their invaluable contributions and insightful discussions, which were instrumental to the success of this research.

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

# A Supplementary materials

## A.1 OGB taxonomy of graph datasets

Graph datasets are classified according to their task, domain, and scale. The task is at the node-, link-, or graph- level; the scale is small, medium, or large; and the domain is nature, society, or information. Our dataset comprises a collection of power grid datasets designed for graph-level tasks, and their size ranges from small to medium [51]. The Open Graph Benchmark [18] contains a diverse set of various sizes and operational specifics of real-world datasets. It contains medium to large-scale datasets that can be used to feed data-hungry models like GNN. For node and link property prediction tasks, OGB has datasets in all domains, *i.e.*, nature, society, and information. However, Table 7 shows the absence of graph datasets in the society domain. We propose PowerGraph, the first collection of real datasets in the *society* domain to fill this gap.

Table 7: OGB taxonomy for graph datasets.

| Domain | Property prediction task | | |
|---|---|---|---|
| | **Node** | **Link** | **Graph** |
| **Nature** | proteins | ddi,ppa | molhiv,molpcba/ppa |
| **Society** | arxiv,products,papers100M | biokg,wikikg2 | - |
| **Information** | mag | collab,citation2 | code2 |

## A.2 The power flow and optimal power flow problem

Planning the operation of a power system involves conducting studies to ensure that power plants produce enough electricity to meet demand while maintaining stable voltage and frequency levels across the network. These studies determine voltage, current, power, and power factor or reactive power at various points in the electric network under normal conditions. They are based on the power flow equations, which model the relationships between voltages, currents, power injections, and loads in the system based on Kirchhoff's and Ohm's laws [52, 53]

### A.2.1 AC Power flow

AC power flow is an integral method for assessing the steady-state operations of the electric power grid, providing insights into the branch loading and bus voltages. It is a nonlinear problem and computationally challenging, especially for large networks. The AC power flow identifies:

- Power : $S = P + jQ$:
    - Active power $P$ [W], which can be power generated $P_g$ or power demanded $P_d$
    - Reactive power $Q$ [VAr], which can be power generated $Q_g$ or power demanded $Q_d$
- Voltage (in polar coordinates) : $|V| \exp j\theta$

To solve for those quantities, the power balance equations for the real and imaginary parts of the power are defined for a single bus as:

$$P_{g_i} - P_{d_i} = \sum_{k=1}^{N} p_{ik}$$
$$Q_{g_i} - Q_{d_i} = \sum_{k=1}^{N} q_{ik} \tag{1}$$

$p_{ij}$ and $q_{ij}$ are the real and reactive power flows in the transmission line or transformers connected to node I. Transmission lines and transformers are characterized by admittance $Y_{ij} = G_{ij} + jB_{ij}$; where $G_{ij}$ [$\Omega$] is the conductance and $B_{ij}$ [$S$] the susceptance. The power flows between buses $i$ and $j$ are given:

$$\begin{cases} p_{ij} = G_{ij}(|V_i|^2 - |V_i|_i|V_j|cos(\theta_{ij})) - B_{ij}|V_i|_i|V_j|sin(\theta_{ij}) \\ q_{ij} = B_{ij}(-|V_i|^2 + |V_i|_i|V_j|cos(\theta_{ij})) - G_{ij}|V_i|_i|V_j|sin(\theta_{ij}) \end{cases} \tag{2}$$

Numerical methods such as the Newton-Raphson method [52] are commonly used to solve nonlinear power-flow equations. This iterative technique begins with initial guesses for unknown variables, including voltage magnitudes and angles at the load and generator buses.

The known and unknown variables depend on the type of bus within the system. Buses without connected generators are classified as loads, or PQ buses, with known power demand ($P_d$ and $Q_d$). Unknowns at these buses include the voltage magnitude ($|V|$) and angle ($\theta$). On the other hand, generator buses, or PV buses, have at least one connected generator with known generated active power ($P_g$) and voltage magnitude ($|V|$). At these buses, the unknowns are the reactive power ($Q_g$) and voltage angle ($\theta$). A special case is the slack bus, arbitrarily chosen with a generator, serving as a reference point for power-flow analysis. At the slack bus, the voltage magnitude ($|V|$) and angle ($\theta$) are set, and the generated active power ($P_g$) and reactive power ($Q_g$) are adjusted accordingly. For a visual representation of each bus type's unknown and known variables see Figure 1.

### A.2.2 Optimal power flow

The optimal power flow determines the optimal output of dispatchable generators in the system while satisfying the nodal balance and the generator and grid operating constraints [54]:

$$\text{Objective function:} \quad \min_{P_{G,i}, Q_{G,i} \, i \in V} \sum_{i \in V} C_i(P_{G,i}) \tag{3}$$

$$\text{subject to:} \tag{4}$$

Nodal balances

$$P_{g_i} - P_{d_i} - g_i^{sh} = \sum_{j=1}^{N} p_{ij}$$

$$Q_{g_i} - Q_{d_i} + b_i^{sh} = \sum_{j=1}^{N} q_{ij} \tag{5}$$

Voltage magnitude limits $v_i^{\min} < v_i < v_i^{\max} \quad \forall i \in V$

Active power limits $P_{g,i}^{\min} < P_{g_i} < P_{g,i}^{\max} \quad \forall i \in V$

Reactive power limits $Q_{g,i}^{\min} < Q_{g_i} < Q_{g,i}^{\max} \quad \forall i \in V$

Line limits $(p_{ij})^2 + (q_{ij})^2 \leq S_{ij}^{\max}$

Figure 1 visualizes the variables predicted at load and generator buses.

### A.3 Physics-based model of cascading failures

The established traditional approach for cascading failure analysis is a quasi-steady state model, such as the OPA model [55], the Manchester model [56], and the Cascades model [21]. We employ the established Cascades model [21, 57] for cascading failure simulations to produce the GNN datasets. Indeed, its application to the Western Electricity Coordinating Council (WECC) power grid demonstrates that Cascades can generate a distribution of blackouts that is consistent with the historical blackout data [48]. Cascades is a steady-steady state model with the objective to simulate the power grid response under unplanned failures in the grid. For that purpose, the model simulates the power system's automatic and manual responses after such failures. Initially, all components are in service and the grid has no overloads. The system is in a steady-state operation with the demand supplied by the available generators, which produce power according to AC- optimal power flow (OPF) conditions [58]. The simulation begins with the introduction of single or multiple initial failures. Then, Cascades simulates the post-outage evolution of the power grid, i.e., identifies islands, performs frequency control, under-frequency load shedding, under-voltage load shedding, AC power flows, checks for overloads, and disconnects overloaded components. The model returns two main results: the demand not served (DNS) in MW and the number of branches tripped after the initial triggering failure. The simulation is performed for a set of power demands sampled from a yearly load curve. An equal number of loading conditions are randomly sampled for each season of the year. We use a Monte-Carlo simulation to probabilistically generate outages of transmission branches (lines and transformers). We define the number of loading conditions and the size of the outage list. Therefore, we are able to simulate a large number of scenarios and thus create large datasets. Each scenario generated is a power grid state and becomes an instance of the dataset. For each combination of loading condition and element in the outage list, we simulate the cascading failure, identify the terminal state of the power grid, quantify the demand not served, and list the tripped elements. Figure 6 shows the structure of the Cascades model [59].

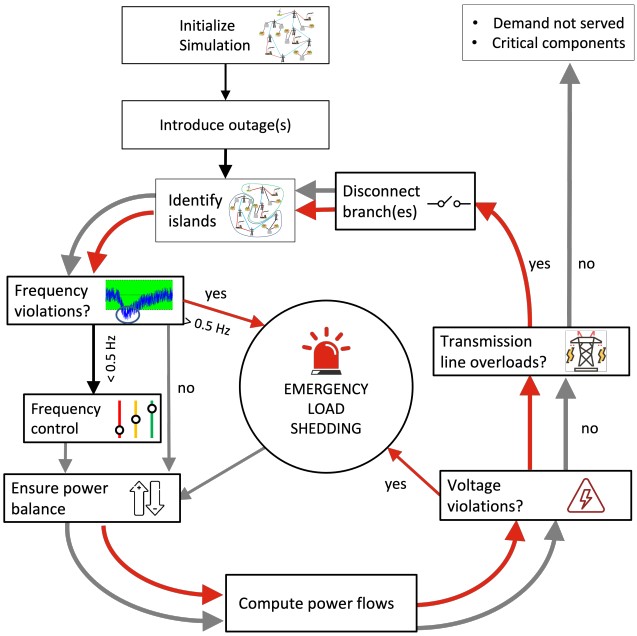

Figure 6: Workflow of the Cascades [59] model, used to simulate cascading failures in power grids. Separate Cascades runs are performed for the different test power grids, namely, IEEE24, IEEE39, UK, and IEEE118.

Table 8: Parameters of the AC physics-based cascading failure model for the selected four test power grids.

| Power system | No. Loading conditions - $n_{loadcond}$ | No. Outage lists $n_{outlists}$ |
|---|---|---|
| IEEE24 | 500 | 43 |
| IEEE39 | 500 | 56 |
| IEEE118 | 500 | 128 |
| UK | 500 | 245 |

## A.4  Class targeted explanations

For benchmarking explanations in section 4, we focus on explaining Category A graphs of the multi-class problem, i.e., the power grids that fail to serve the demand (DNS>0). The objective is to shed light on the tripped lines after the first contingency. We use the multi-class problem rather than the binary classification problem that classifies states according to the demand not served (DNS) only, i.e. distinguishes power grids that serve the demand (DNS=0, label 1) from the ones that do not (DNS>0, label 0). In the multi-class problem, the model learns to distinguish cascading failure scenarios, while in the binary setting, Category A and B are considered the same type of grids (class DNS>0). Explaining DNS>0 in the multi-class problem allows us to focus on the case where some lines are tripped when DNS>0 and, therefore, expect the model to learn the cascading edges for this class of grids.

## A.5  Reconstructing temporal graphs for in-depth analysis of cascading failures in PowerGraph

Each instance of the graph-level dataset in PowerGraph, designed for cascading failure analysis, is accompanied by an explainability mask that encodes the stages of the cascading failure process. This mask is stored in a MATLAB structure exp.mat, see in Appendix C.1. Each entry in the MATLAB structure is a vector containing all failures at all stages, with the order of the vector corresponding to the sequence of line failures in the cascading event. Using this information, a temporal graph can be reconstructed from PowerGraph.

## A.6  Balanced accuracy

**Definition**  The balanced accuracy is the arithmetic mean of the specificity and the sensitivity. The sensitivity or true positive rate or recall measures the proportion of real positives that are correctly predicted out of all positive predictions that could be made by the model. The specificity or true negative rate measures the

proportion of correctly identified negatives over the total negative predictions that could be made by the model. The balanced accuracy is then expressed as:

$$\text{Balanced Accuracy} = \frac{\text{Sensitivity} + \text{Specificity}}{2} = \frac{1}{2} \cdot \left( \frac{TP}{TP + FN} + \frac{TN}{TN + FP} \right) \tag{6}$$

The balanced accuracy has the advantage of accounting for imbalance in the explanatory mask. In the context of cascading failure detection, we know that most of the components (links) in the grid will not fail. Therefore, the edge mask has many values that are zeros and only a few that are ones. The balanced accuracy measures if the method was able to recognize both failing and not failing edges, while giving the same importance to both detections.

## B  Explainability methods

To explain the decisions made by the GNN models, we adopt different classes of explainers, including generative and non-generative methods. We detail properties and additional results regarding explainability methods run on PowerGraph datasets in this section.

### B.1  XAI properties

**Model-aware**. Gradient-based methods compute the gradients of target prediction with respect to input features by back-propagation. Features-based methods map the hidden features to the input space via interpolation to measure important scores. Decomposition methods measure the importance of input features by distributing the prediction scores to the input space in a back-propagation manner.

**Model-agnostic**. Perturbation-based methods use a masking strategy in the input space to perturb the input. Surrogate models use node/edge dropping, BFS sampling, and node feature perturbation. Counterfactual methods generate counterfactual explanations by searching for a close possible world using adversarial perturbation techniques [60].

### B.2  Non-generative explainability methods

In our experiments, we compare the following methods: **Random** gives every edge and node feature a random value between 0 and 1; **Saliency (SA)** measures node importance as the weight on every node after computing the gradient of the output with respect to node features [39]; **Integrated Gradient (IG)** avoids the saturation problem of the gradient-based method Saliency by accumulating gradients over the path from a baseline input (zero-vector) and the input at hand [40]; **Grad-CAM** is a generalization of class activation maps (CAM) [41]; **Occlusion** attributes the importance of an edge as the difference of the model initial prediction on the graph after removing this edge [38]; **GNNExplainer (E,NF)** computes the importance of graph entities (node/edge/node feature) using the mutual information [42]; We also use **GNNExplainer** that considers only edge importance; **PGM-Explainer** perturbs the input and uses probabilistic graphical models to find the dependencies between the nodes and the output [43]; **SubgraphX** explores possible explanatory sub-graphs with Monte Carlo Tree Search and assigns them a score using the Shapley value [44].

### B.3  Generative explainability methods

Non-generative explainability methods like gradient-based or perturbation-based methods optimize individual explanations for a given instance. They lack a global understanding of the whole dataset or the ability to generalize to new unseen instances. Non-generative methods have been developed to tackle this problem. They learn the initial data distribution before generating individual explanations. Therefore, generative methods learn the underlying distributions of the explanatory graphs across the entire dataset, providing a more holistic approach to GNN explanations. **GSAT** [45] creates inherently interpretable and generalizable GNNs by constraining information flow from the input graph to the prediction explainability method by treating attention as an Information Bottleneck (IB) and introducing stochasticity into the attention process. The reinforcement learning-based xAI method **RCExplainer** [47] defines explanation as a sequential decision process where salient edges are added to construct an explanatory subgraph, guided by a policy network that predicts edge additions based on their causal effect on the prediction and their collaborative impact with previously added edges. Using a diffusion-based process, **D4Explainer** [46] integrates generative graph distribution learning into its optimization objective, enabling the generation of diverse counterfactual graphs within the distribution and identifying discriminative graph patterns that explain specific class predictions, serving as model-level explanations.

Table 9: Explainability methods tested on the PowerGraph benchmark.

| Explainer | Model-aware/agnostic | Target | Type | Flow |
|---|---|---|---|---|
| SA | Model-aware | N/E | Gradient | Backward |
| IG | Model-aware | N/E | Gradient | Backward |
| Grad-CAM | Model-aware | N | Gradient | Backward |
| Occlusion | Model-agnostic | N/E | Perturbation | Forward |
| GNNExplainer | Model-agnostic | N/E/NF | Perturbation | Forward |
| PGM-Explainer | Model-agnostic | N/E | Perturbation | Forward |
| SubgraphX | Model-agnostic | N/E | Perturbation | Forward |
| GSAT | Model-agnostic | E | Mask Generation | Factual |
| RCExplainer | Model-agnostic | SUBGRAPH | RL-MDP | Factual |
| D4Explainer | Model-agnostic | E | Diffusion | Counterfactual |

## B.4 The *GraphFramEx* evaluation framework

We construct the evaluation pipeline of the explainability methods following GRAPHFRAMEX, a systematic framework for evaluating explainability methods in the context of graph classification. We consider three aspects of *users' needs* in our evaluation protocol, namely explanation focus, mask nature and mask transformation.

**Focus of explanation.** Some users want to explain why a certain decision has been returned for a particular input. In this case, the task of explaining has a more applied nature: they are interested in the *phenomenon* itself and try to reveal findings in the data, i.e. explain the true labeling of the nodes. The model's predictions are ignored in the explanation process. Others prefer the behavior of GNN *model* and try to explain the logic behind the model, i.e.,textitmodel behavior and try to explain the logic behind the model, i.e., the predicted labels. These equally complementary and important reasons demand different analysis methods. The choice of explanation focus determines the explanation objective and evaluation.

**Mask sparsity** Because there is no such thing as a "good" size for an explanation, it is even harder to compare explainability methods. Existing explainability methods return different sizes of explanations by default. GraphFramEx defines three strategies to reduce explanation size: sparsity, threshold, and topk, which transform the edge mask $M$ into a sparser version $M_t$. We decide to use the topk strategy because it is the only strategy that enforces a maximum number $k$ of edges independently of the size of the graph and the explainer methodology. This independence property is important as human-intelligible explanations cannot exceed a certain number of graph entities. Too small explanations omit important elements and will not be sufficient, while too big explanations contain irrelevant nodes and edges and will not be necessary.

**Masking strategies** To convert importance attributions, i.e., the explanatory mask, to an explanatory subgraph, one must find a strategy to assign those importance scores to the graph entities. There are two strategies: the *hard* masking and the *soft* masking. Let's say that, for sparsity reasons, we want to keep only the $topk \in \mathcal{K}$ edges in a graph, i.e., the $k$ edges that have the highest explanatory attribution scores, in the final explanations. The hard masking function $\chi_H : \mathcal{G} \times \mathcal{K} \to \mathcal{G}$ picks the $topk$ edges from a graph $G$ so that the number of edges and nodes is reduced. Applying the hard selector on an explanation $G' = h(G)$ for the $topk$ edges, we obtain a *hard* explanation $\chi_H(G', t) = G'(\mathcal{V}', \mathcal{E}', \mathbf{X}', \mathbf{E}')$, such that $\mathcal{V}' \subseteq \mathcal{V}$ and $\mathbf{X}' = \{X_j \| v_j \in \mathcal{V}'\}$, where $v_j$ and $X_j$ denote the graph node and the corresponding node features. The hard explanation has only the nodes connected to the remaining important edges and is usually smaller than the input graphs that very likely do not lie in the initial data distribution. The soft masking function $\chi_S : \mathcal{G} \times \mathcal{K} \to \mathcal{G}$ instead sets edge weights to zero when edges are to be removed. Therefore it preserves the whole graph structure with all nodes and edge indices. Given an explanation $G'$ and the $topk$-sparse explanatory mask $M_t$ that keeps the $topk\%$ highest values in $M$ and sets the rest to zero, we can express the *soft* explanation at $topk$ as $\chi_S(G', t) = G'(\mathcal{V}, \mathcal{E}, \mathbf{X}, \mathbf{E}, M_t)$. It has a similar edge index and nodes as the input graph $G$, but unimportant edges receive zero weights. Note here that, unlike the soft masking strategy, the hard masking strategy might break the connectivity of the input graphs, resulting in explanations represented by multiple disconnected subgraphs.

For PowerGraph, we compare explainability methods taking a *phenomenon* focus, i.e. explaining the true class label Category A. We use the $topk$ strategy to get sparse and intelligible explanations. Masks are *soft* on the edges, i.e., we assign weights to the important edges. Explanations are weighted explanatory subgraphs, where edges are given importance based on their contribution to the true prediction in the multi-class setting. Figure 5 reports the fidelity+ scores for the four power grid datasets.

## B.5 Faithfulness metric

To measure the faithfulness of the explanations, we use either the fidelity- or the fidelity+ scores defined in [61, 37]. We evaluate the contribution of the produced explanatory subgraph to the initial prediction, either by giving only the subgraph as input to the model (fidelity-) or by removing it from the entire graph and re-running the model (fidelity+). As explained in section A.4, the generated explanations in the context of PowerGraph are the tripped lines and, therefore, should be necessary but not sufficient for the grid class. Indeed, the subgraph resulting from isolating the cascading branches does not represent a power grid. Therefore, fidelity- is not relevant in the context of the PowerGraph benchmark, and we evaluate the faithfulness of explanations using the fidelity+ metric defined in equations 7 and 8. The fidelity score can be expressed either with probabilities ($fid_+^{prob}$) or indicator functions ($fid_+^{acc}$). We adopt the $fid_+^{prob}$, as it is more precise for our classification problem. $f$ is a pre-trained classifier. We denote by $\hat{y}_i^{G_C}$ and $\hat{y}_i^{G_{C\setminus S}}$ the model's predictions when taking as input respectively the input graph $G_C$ and its complement or masked-out graph $G_{C\setminus S}$.

$$fid+^{acc} = \frac{1}{N}\sum_{i=1}^{N}\|\mathbb{1}(\hat{y}_i^{G_C} = y_i) - \mathbb{1}(\hat{y}_i^{G_{C\setminus S}} = y_i)\| \tag{7}$$

$$fid+^{prob} = \frac{1}{N}\sum_{i=1}^{N}(f(G_C)_{y_i} - f(G_{C\setminus S})_{y_i}) \tag{8}$$

While the GraphFramEx framework distinguishes two types of explanations, according to whether they are *necessary* or *sufficient*, we only consider necessary explanations.

## B.6 Additional XAI results

This section presents additional findings regarding the faithfulness of generated explanations across different sparsity levels. We manipulate the number of retained top-k edges, ranging from 1 to 20 edges, beginning with the most important edge. Figures 7 and 8 demonstrate the superiority of gradient-based methods such as Saliency and IntegratedGrad, Occlusion, and GSAT. Notably, we observe that the fidelity computed on the accuracy of the classification ($fid+^{acc}$) shows more fluctuating trends compared to the smoother curves of the fidelity metric computed with probabilities ($fid+^{prob}$), aligning with their respective definitions given in equations 7 and 8.

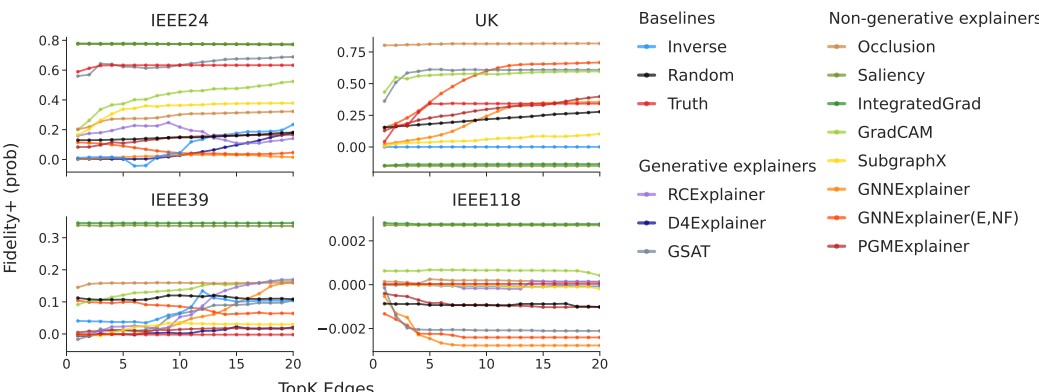

Figure 7: Faithfulness computed with $fid+^{prob}$ defined in section B.5 when varying the number of $topk$ explanatory edges from 1 to 20.

## C   Access to PowerGraph Dataset

### C.1   Dataset documentation and intended uses

PowerGraph is the collection of the following GNN datasets: UK, IEEE24, IEEE39, IEEE118 power grids. We use `InMemoryDataset` [62] class of Pytorch Geometric, which processes the raw data obtained from the Cascades [63] simulation. For each dataset UK, IEEE24, IEEE39, and IEEE118, we provide a folder containing the raw data organized in the following files for node-level tasks, i.e., power flow and optimal power flow analyses:

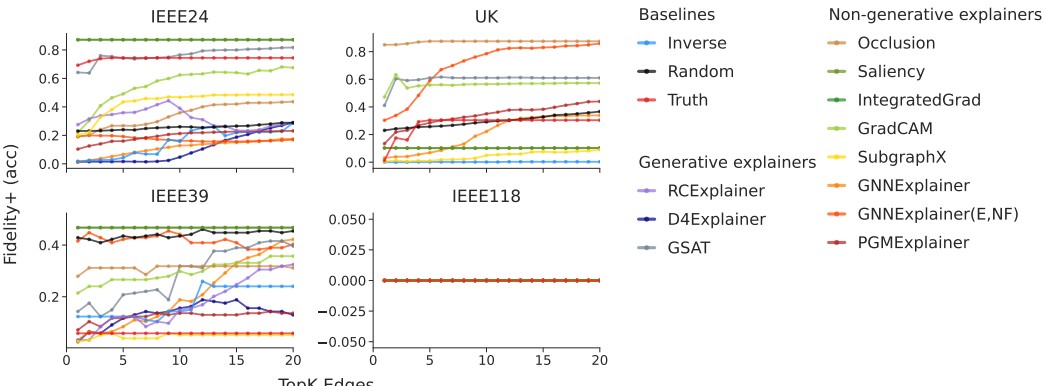

Figure 8: Faithfulness computed with $fid+^{acc}$ defined in section B.5 when varying the number of $topk$ explanatory edges from 1 to 20.

- `edge_attr.mat`: edge feature matrix for the power flow problem (branch conductance $G_{ij}$, branch susceptance $B_{ij}$.)

- `edge_attr_opf.mat`: edge feature matrix for the optimal power flow problem (branch conductance $G_{ij}$, branch susceptance $B_{ij}$.)

- `edge_index.mat`: list of branches, represented as connection from node - to node.

- `edge_index_opf.mat`: list of branches, represented as connection from node - to node.

- `X.mat`: node feature matrix for the power flow problem (active power generation $P_g$ - active power demand $P_d$, reactive power generation $Q_g$ - reactive power demand $Q_d$, voltage magnitude $V$, and voltage angle $\theta$, the number of loads $N_{loads}$, and number of generators $N_{gen}$).

- `Xopf.mat`:node feature matrix for the optimal power flow problem (active power generation $P_g$ - active power demand $P_d$, reactive power generation $Q_g$ - reactive power demand $Q_d$, voltage magnitude $V$, and voltage angle $\theta$, the number of loads $N_{loads}$, and number of generators $N_{gen}$).

- `Y_polar.mat`: node output matrix for the power flow problem (active power generation $P_g$ - active power demand $P_d$, reactive power generation $Q_g$ - reactive power demand $Q_d$, voltage magnitude $V$, and voltage angle $\theta$).

- `Y_polar_opf.mat`: node output matrix for the optimal power flow problem (active power generation $P_g$ - active power demand $P_d$, reactive power generation $Q_g$ - reactive power demand $Q_d$, voltage magnitude $V$, and voltage angle $\theta$).

For graph-level tasks, i.e., cascading failure analysis:

- `blist.mat`: list of branches, represented as connection from node - to node

- `of_bi.mat`: binary classification labels ($DNS = 0$ or $DNS \neq 0$)

- `of_reg.mat`: regression labels ($DNS$)

- `of_mc.mat`: multi-class classification labels (See Table 2)

- `Bf.mat`: node feature matrix (Net active power at bus $P_{net}$, Net apparent power at bus $S_{net}$, Voltage magnitude $V$

- `Ef.mat`: edge feature matrix (Active power flow $P_{i,j}$, Reactive power flow $Q_{i,j}$, Line reactance $X_{i,j}$, Line rating $lr_{i,j}$)

- `exp.mat`: ground-truth explanation (See A.5)

## C.2 Download Dataset

The dataset can be viewed and downloaded by the reviewers from https://figshare.com/articles/dataset/PowerGraph/22820534 (node-level ~1.08GB and graph-level ~2.7GB, when uncompressed):

Node-level data:

```bash
#!/bin/bash
wget -O data.tar.gz "https://figshare.com/ndownloader/files/46619152"
tar -xf data.tar.gz
```

Graph-level data:

```bash
#!/bin/bash
wget -O data.tar.gz "https://figshare.com/ndownloader/files/46619158"
tar -xf data.tar.gz
```

## C.3 Author statement

The authors state here that they bear all responsibility in case of violation of rights, etc., and confirm that this work is licensed under the CC BY 4.0 license.

## C.4 Hosting, Licensing, and Maintenance Plan

The code to obtain the PowerGraph dataset in the `InMemoryDataset` [62] format and to benchmark GNN and explainability methods is available as a public GitHub organization at `https://github.com/PowerGraph-Datasets/`. The authors are responsible for updating the code in case issues are raised and maintaining the datasets. We aim to extend the PowerGraph with new datasets and include additional power grid analyses, including solutions to the unit commitment problem. Over time, we plan to release new versions of the datasets and provide updates to the results for both the GNN accuracy and the explainability analysis. In addition, the code will be updated if new pytorch/torch-geometric versions are released or crucial python packages are updated. The data is hosted on figshare at `https://figshare.com/articles/dataset/PowerGraph/22820534`. The authors give public free access to the PowerGraph dataset. The datasets are identified with the `DOI:10.6084/m9.figshare.22820534`. The work in this paper (code, data) is licensed under the CC BY 4.0 license.

## C.5 Code Implementation

We run a hyper-parameters grid search over different GNN models, using torch-geometric 2.3.0 [62] and Torch 2.0.0 with CUDA version 11.8 [64, 65]. The benchmark node and graph classification and regression models experiments are performed on the GPU nodes of the ETH Euler clusters [66]. For the explainability analysis, experiments are conducted on 8 AMD EPYC 7742 CPUs with a memory of 5GB each on the ETH Euler clusters [66].

