# OpenReview forum: "PowerGraph: A power grid benchmark dataset for graph neural networks"
_NeurIPS.cc/2024/Datasets_and_Benchmarks_Track — NeurIPS 2024 Track Datasets and Benchmarks Poster_

### Official Review · Reviewer_j6SE · 2024-07-19
**Review of submission 1725**

**Rating:** 8
**Confidence:** 4
**Correctness:** Looks good. The author offers a real …
**Clarity:** Fair.

**Review:**

**Pros**
1. This paper comprises and releases a new dataset named PowerGraph, specifically designed for GNN applications in power grid analysis.
2. Based on the new dataset, this paper conducts extensive experiments to evaluate the performance of various GNN architectures on both node-level and graph-level tasks.
3. The dataset includes ground-truth explanations for cascading failure analysis, providing a valuable resource for developing real explainable methods for power grid systems.

**Cons**
1. The paper lacks a clear articulation of the gaps or limitations in existing power grid datasets that PowerGraph aims to address. Many widely used systems, such as IEEE-118, have been extensively studied, and the specific improvements or unique contributions of PowerGraph over these existing datasets should be more explicitly stated.
2. The authors claim to provide a new dataset for power grid analysis, but the work appears more akin to a library.
3. The authors state they contribute to the OGB library, but there is no related information in the official documentation. If my search is not comprehensive, please provide the concrete link.
4. The authors' performance analysis of the new dataset is insufficient, showing only the best results. Other results should also be included in the appendix.
5. The innovation in explainability introduced in the dataset is limited, despite using various XAI methods. In the era of large language models, textual explanations might be more meaningful for power grid management.

**Strengths:**

See the previous content.

**Additional Feedback:**

Not applicable.

**Documentation:**

Yes. Looks comprehensive.

**Ethics:**

Not applicable.

**Limitations:**

Not applicable.

**Opportunities For Improvement:**

The authors should clearly articulate the problem related to the absence of suitable datasets for GNN applications in power grids. They should highlight the limitations of existing datasets, such as IEEE-118, and explain how PowerGraph addresses these issues.

**Relation To Prior Work:**

Yes, the discussion of related work is adequate and prior work is cited when appropriate. However, the previous work on existing four power grid systems should be further introduced.

**Summary And Contributions:**

The paper introduces "PowerGraph," a comprehensive benchmark dataset tailored for Graph Neural Networks (GNNs) in power grid applications. PowerGraph aims to address the lack of publicly available graph datasets specifically designed for training and benchmarking machine learning models in electrical power grid applications. It includes datasets for power flow, optimal power flow, and cascading failure analyses of power grids. The paper benchmarks various GNN methods for node-level and graph-level tasks and provides ground-truth explanations for cascading failure analysis.

The main contributions are summarized as follows:
1. Introduction of the "PowerGraph" dataset, encompassing IEEE24, IEEE39, IEEE118, and UK transmission systems, tailored for power flow, optimal power flow, and cascading failure analysis.
2. A comprehensive benchmarking of GNN architectures for node-level and graph-level tasks, evaluating performance on power flow, optimal power flow, and cascading failure scenarios.
3. Provision of ground-truth explanations for cascading failures, enhancing the interpretability of GNN models in power grid applications.
4. Release of all data and code, facilitating further research and development in the field of GNN-based power system modeling.

---

> ### Author Rebuttal · Authors · 2024-08-16
>
> •	Thank you for your suggestion. Please refer to the research gaps and contributions listed in the introduction, respectively:
> “In this work, we present PowerGraph, aiming to address the following research gaps:
> • A public dataset designed for various power systems to solve power flow and optimal power flow problems using GNN supervised learning techniques.
> • A public dataset encompassing a wide range of cascading failure scenarios for different graph-level tasks.
> • A real-world dataset for GNN graph-level tasks with clear ground-truth explanations for GNN explainability.
> • A dataset that contributes to the society domain within the OGB graph dataset taxonomy [18].”
> “Our contributions are as follows:
> • Introducing a data-driven approach for analyzing power flow and cascading failure events in power grids in real-time.
> • Assessing and benchmarking various GNN architectures and hyperparameters.
> • Providing the first real-world GNN dataset with empirical explanations to benchmark GNN explainability methods.
> • Making the dataset easily accessible in a user-friendly format, allowing the GNN community to
> experiment with different architectures for node and graph-level applications.
> • Offering a range of tasks at both the node and graph levels, including regression, binary classification, and multi-class classification.”
> Furthermore, to address your comment, we have added a Limitations section before the conclusions.
> • To better address your concerns, could you please elaborate on which aspects of our work seem more akin to a library? This will help us refine our presentation and clarify the primary contribution of our dataset for power grid analysis in the revised paper. Your guidance is much appreciated.
> •	Thank you for your observation. We have indeed applied for our dataset to be included in the OGB library. However, the review and inclusion process are currently taking longer than anticipated. The acceptance is contingent upon our work being presented at a conference, which is a prerequisite for listing in the OGB library. We will update the documentation with the official link as soon as our dataset is officially included. We appreciate your patience and understanding in this matter.
> To address the comment, we have changed the statement within our research gap in the Introduction, respectively:
> “A dataset that contributes to the society domain within the OGB graph dataset taxonomy [18].”
>
> • To address your request, we have presented the complete results, including all models and their performances, on our website https://powergraph.ivia.ch. This approach allows us to avoid lengthy tables in the appendix while still providing comprehensive data for those interested. We hope this solution meets your expectations for detailed information.
> To address your comment, we have added a clarification to the observation paragraph of Section 3:
> “In addition to the presented result in this paper and its appendices, we present the complete results, including all models and their performances, on our website: https://powergraph.ivia.ch.”
> •We appreciate the reviewer's insightful comment. In our work with graph structures and GNNs, we utilize explainability methods specifically adapted for graph data. These methods effectively consider both the connectivity within the graph (including edges and edge features) and the characteristics of each node (node features). After generating the explanatory graph, it is possible to transform this subgraph into a text-based explanation using a transformer model. However, we have not pursued this approach because our primary users are power grid experts, for whom a graphical explanation—a subgraph of the grid—offers more intuitive interpretability. Nevertheless, to accommodate machine learning experts who may not be familiar with transmission grids, we plan to introduce a post-processing verbalization of the graph explanations.
> We have added the following clarification to the Experimental setting and evaluation metrics paragraph of Section 4:
> “The explanations are subsets of the grids. We prefer generating graph-based explanations over textual ones, as graph structures are more intuitive and effective for conveying critical nodes and edges to experts in transmission grids.”
> •To address your comment, we have added a Limitation section before the conclusions.
> While this study provides a valuable benchmark for power grid analysis using GNN models, there are limitations that should be acknowledged. Using power grid models, such as the IEEE test case systems, is a standard practice in power engineering modeling and analyses. Although these power systems are not ideal, they are still based on real power grids. The main reason we utilize these grids in our study is because real power system data is safety-critical and often classified, making it inaccessible for research. Furthermore, cascading failures are rare events, and data on their progression and the conditions under which they occur are largely unavailable. As a result, synthetic data is the only viable option for cascading failure analysis, which could limit the generalizability of the findings to real-world scenarios. Nevertheless, we use the Cascades model to simulate cascading failures and generate GNN datasets, which have been validated with historical blackout data from the WECC grid [47]. Furthermore, the small number of buses in the IEEE test cases may constrain the ability of the models to capture long-range dependencies, potentially affecting their performance on larger, more complex power grids. While this work demonstrates a method for modeling power systems with GNNs, other approaches could be explored in future research to enhance model accuracy and robustness. For instance, incorporating energy price variations in the optimal power flow analysis could provide a more comprehensive understanding of the economic aspects of power grid operations, which was not considered in this study"

---

> > ### Comment · Reviewer_j6SE · 2024-08-17
> > **Thanks for your response**
> >
> > I appreciate the author making such a detailed response. Although the response format is a little hard to read, I still read it completely.
> >
> > The author has addressed most of my concerns. I just have one question now: Does the author make a lot of simulation results (e.g., cascading failure events) on existing power systems, such as IEEE-118, which have not been done by previous power datasets?
> >
> > If yes, I would increase my score accordingly.

---

> > > ### Author Response · Authors · 2024-08-18
> > > **Answer to Reviewer j6SE**
> > >
> > > Thank you for your feedback, and we apologize for any difficulty in reading our response. The 6000-character word limit constrained our ability to present the information in a more reader-friendly format. We sincerely appreciate your effort in reading through it despite this challenge.
> > >
> > > To address your question, yes, to our knowledge, we are the first to provide simulation results and benchmark multiple models on power systems like the IEEE-118, with a particular focus on cascading failures, which has not been explored in previous power datasets. We are also extending this analysis to larger grids, such as the 2000-bus Texas system.

---

> > ### Comment · Reviewer_j6SE · 2024-08-18
> >
> > Thanks for your fast response! I am satisfied with the author's response! Concretely, I think the cascading failure scenarios in power grids are highly similar to those in communication networks to some extent, such as the complexity and diversity of failure instances. Simulating cascading failures is necessary and important to help design more powerful models for power grids, especially for intelligent fault diagnosis systems. Therefore, I have revised my scores and recommend accepting this paper!

---

> > > ### Author Response · Authors · 2024-08-23
> > > **Reply to Official Comment**
> > >
> > > We sincerely appreciate your valuable comments and support. Thank you!

---

### Official Review · Reviewer_qprq · 2024-07-22
**New and highly relevant dataset with room for improving the presentation**

**Rating:** 6
**Confidence:** 4
**Correctness:** I have no reason to doubt the correct…

**Review:**

The work is original and highly significant for researchers working in the intersection of power grids and ML. All listed problems power flow, optimal power flow and cascading failures are all relevant topics. There is a need for high quality power grid datasets. The real-world explanations for XAI methods can be very helpful when developing and benchmarking new XAI methods. The quality of the experiments seem to be sound and of appropriate quality.


### Pro
* highly relevant field, high quality benchmarks for power grid data are missing
* the datasets deal with relevant problems (power flow, optimal power flow and cascading failures)
* real word XAI explanations are helpful

### Cons
* presentation can be improved (see Opportunities for Improvement)
* the quality of the results should be assessed with respect to real world applications. Are the obtained MSE good enough to be of practical usage for grid operators?
* The used power grids are relatively small, it would be great if at least one large power grid could be added, e.g. Synthetic power grid of Texas by Birchfield et al.
* Baselines, such as gradient boosted trees, are missing, which would be helpful to assess the advantages of GNNs.

**Strengths:**

The work is highly relevant, those datasets can be helpful to develop new GNN algorithms used for power grids, since GNNs can be very helpful in the context of power grids, but there is a lack of appropriate benchmark datasets. Overall, the paper has a clear structure.

**Additional Feedback:**

### Questions
> The known input quantities are reported for different node types, and the unknown quantities are set to zero (indicated by an empty cell in the picture)

Why is 0 imputation used? Couldn't that cause problem when the real "known" values might be 0 or close to 0?

> Figure 3 reveals that the PF models do not
necessarily outperform the OPF models in predicting voltage magnitude and angle, particularly for the IEEE118-bus system

Is this really a statement about the model? Are PF models different from OPF models? I guess the term model denotes ML model, but to me that statement looks more to be connected to the task, than the models. Also in the following, I find it confusing to talk about PF and OPF models that perform better or worse. One could think that models are trained on OPF and then evaluated for PF and perform better or worse.


I am missing a discussion of the quality of the models for section 3 and 4. Are those MSE good or bad? Sufficient for practical use? What other methods are currently used? ML-based or simply simulations or any simplified models? What is the benchmark outside of ML that ML needs to compete with? What levels of performance would we need for real world applications?

> To benchmark explainability methods, having the best GNN model is not essential

Why not? Wouldn't better models decrease the risk of looking at "wrong" explanations?

- Why are small numbers of MPL sufficient? Ringsquandl et al. [Ring2021] show that deep GNNs are needed in case of power grids. What is the difference? I also thought that cascading failures are often not local phenomena, but rather propagate through large parts of the networks and CFs are in reality often caused across large distances.

[Ring2021] https://doi.org/10.1145/3459637.3482464


### Further remarks
> Therefore, we must develop grid analysis algorithms to ensure reliable operations

The "must" might be a bit strong, I agree that grid analysis algorithms for ensuring reliable operations are beneficial, but there might be alternatives to this.

>  However, the R [ 37] score for the remaining power systems remains below 0.26. Ideally, we aim for an R score
Typo, R2 missing (2)

- Table 2: The check mark showing cascading failures was a bit counterintuitive for me, since that are usually states that should be avoided, maybe simple wording such as: cf occurs, no cf might be easier to read.

> Benchmarking power flow analysis: node-level tasks

I was surprised to read something about OPF in the discussion paragraph, in the introduction it seemed like there are three types of tasks, PF, OPF and CF. Maybe the section could be renamed if it also includes OPF.

> Among these, only GCN does not consider edge features, resulting in lower performance in most cases

That is clearly a difference to the power flow tasks and should be explained. I assume that line limits are reached and provided as a edge feature and hence important parameters for potential cascades?


#### Small power grids
The small size of the investigated power grids could be the main reason for the discrepancy to the observation by Ring[2021], which would question the generalizability of the dataset and potential usage.

**Clarity:**

Overall, the paper is organized and well-written. Looking at Opportunities For Improvement, the presentation can and should be improved.

**Documentation:**

The usage of the dataset is documented on GitHub.

**Ethics:**

I do not have ethical concerns.

**Limitations:**

A dedicated section could be added. There are obvious limitations, such as using power grid models (IEEE test cases), but using those models is state-of-the art. Nevertheless, listing them could be beneficial for future work. Furthermore, the small number of buses could be a limitation to benchmark models for long-range dependencies.

**Opportunities For Improvement:**

* The structure in the result sections can be improved. The first sentences in 3 Benchmarking power flow analysis: node-level tasks and 4 Benchmarking cascading failure analysis - graph-level tasks: classification, regression, and explainability are basically the same, so not really helpful to distinguish those sections. Maybe the same models do not have to be introduced twice, maybe the leading sentences of the paragraphs can give an overview of what to expect or show the key message instead of this redundant information. This also continues regarding the split of the datasets, number of epochs and so on. If it is the same, it could be combined, maybe even moved to the appendix, if it is only needed for reproducibility and important to follow the story line.
* The discussion of 3 and 4 could also be combined to show similarities and differences and explain reasons for both. Otherwise, it is really redundant again, and the reader has to look for discrepancies.

* Sometimes, I am wondering about the level of details, for example "graph connectivity information is encoded in COO format". Such information can be moved to the appendix, the main section should focus on information needed to follow the storyline.
* The size of the investigated datasets might be too small to adequately model relevant phenomena, it would be great, if the authors add larger grids, such as the 2000 bus Texan grid by Birchfield et al. (https://electricgrids.engr.tamu.edu/electric-grid-test-cases/activsg2000/)
* Baselines, e.g. Gradient Boosted Trees are missing
* The tables 4 and 5 only show the results of the best-performing models. Can the full tables with all models and their performances be added to the appendix?

**Relation To Prior Work:**

Other publicly available datasets are named in the introduction.

**Summary And Contributions:**

The paper introduces new power grid datasets and highlight the relevance of power grids for society and the need to have high quality datasets to develop appropriate ML models. The introduced datasets are tailored for graph neural networks, and the authors aim to close a gap of missing datasets in the field of power grids. The datasets are actually three different datasets, dealing with power flow, optimal power flow and cascading failures. Lastly, the dataset also contains real-world explanations, which can be helpful when developing XAI methods in the context of power grids.

---

> ### Author Rebuttal · Authors · 2024-08-16
>
> •	Thank you very much for your constructive suggestions. We have created a unique combined section, “4) Benchmarking the PowerGraph dataset”, while explainability methods are discussed in the section “5) Benchmarking explanations on the graph-classification models”, as per the reviewer’s comment.
>
> In the combined Section 4, the experimental setting, which had many similarities, is now placed in a separate paragraph, “Experimental settings and evaluation metrics”. The results discussion on both the power flow (node-level) and the cascading failure analysis (graph-level) is placed in a unified ‘Discussion’ paragraph.
>
> •	We have removed the redundant details, such as the "graph connectivity information encoded in COO format”, as you rightfully pointed out. This information is redundant as it is a technicality and the most established way to handle graph structures nowadays for GNN.
>
> •Thank you for suggesting to include larger grids, such as the 2000-bus Texan grid by Birchfield et al. We agree that incorporating larger datasets could enhance the modeling of relevant phenomena. We included this dataset in the format of our existing datasets in our work for optimal power flow, power flow, and cascading failure analysis.However, due to time constraints, we are not able to run the full grid search of experiments for node-level tasks and graph-level tasks on our resources. Furthermore, our initial research on the dataset reveals that for many instances, for the assumed load profile, the AC OPF and PF do not converge. However, we have set several experiments to further test the data for the instances where convergence is ensured. These experiments on this larger dataset will unlikely be completed before the final deadline, but the results will be shown at the conference upon paper acceptance and on the website powergraph.ivia.ch as soon as ready. We believe that including this dataset, even without exhaustive experimentation, will still provide additional data for researchers to train GNN models and set the stage for further research. We appreciate your understanding and look forward to extending our analysis with this dataset near future.
> To address your comment, we have added the following to the Conclusions paragraph:
>
> "We will also benchmark methods on larger synthetic power systems, such as the grid in [49], to assess whether deeper GNN architectures are necessary, as noted in [50]. A raw dataset is available at https://powergraph.ivia.ch."
>
> •	Although our dataset was specifically designed to test GNN, we also added Gradient Boosted Trees for comparison. We also believe that there should be some space for other models besides GNN. The results are included in the leaderboard on our website https://powergraph.ivia.ch. Furthermore, we have added the following comments in the Observation and Discussion paragraph of Section 3, respectively:
>
> ‘The website also includes Gradient Boosted Trees results as a baseline for comparison.’
>
> ‘We observe the limitations of GNNs, particularly in node regression tasks, where simpler models like Gradient Boosted Trees (GBT) outperform them. However, this trend does not extend to graph-level tasks, where GNNs consistently prove to be the superior method. This highlights the need for further development of GNN models tailored specifically for node regression tasks, critical in power systems analysis. At the same time, even minor improvements in classification tasks should not be underestimated, as they can significantly impact decision-making in critical infrastructure. Therefore, it is crucial to develop highly precise and accurate models to ensure safe power grid operations for system operators.’.The GBT results are attached.
>
> •	We have added a Limitation section before the Conclusions section.
>
> Questions:
>
> •	We opted for zero imputation to maintain the dataset's dimensionality structure. However, we have not observed significant differences in predicting quantities close to zero compared to variables with different orders of magnitude. We plan to explore different imputation methods in our future experiments. We have added the following clarified in the caption of Figure 1.
>
> •	We are referring to the ML models not the PF or OPF, we have changed the sentence as follows:
> “the OPF task is inherently more complex, resulting in a higher mean squared error (MSE) for the ML model trained on it compared to the one trained on the PF task, as illustrated in Figure 3 which presents the node-averaged MSE for each predicted physical quantity.”
>
> •	We have added the following sentence to the Discussion paragraph of Section 3:
>
> "We train our models using data from current simulation methods (see Appendix A2 and A3), aiming for the highest accuracy in classification and regression to replicate those results, while the key advantage of our ML models lies in surpassing traditional solvers or simulations in computational efficiency."
>
> • In our case the model with balanced accuracy above 95% was employed, thus that case wrong explanation are mostly caused by inaccurate explainability methods.
>
> •The reason might be due to the size of our grid and the well-known oversmoothing phenomena in GNN. In the cited reference, they tested GNN on distribution networks, while we focus on Transmission networks, entailing an entirely different topology and properties.
>
> Further remarks
>
> •We modified the sentence in the abstract and changed Table 2.
>
> •We modified section 3, now named Benchmarking PowerGraph, specifying that we deal with the PF, OPF and CF problems.
>
> •Line limits are crucial in cascading failures and thus represent highly relevant data for inclusion in the ML model. We included into the discussion paragraph of Section 3:
> "This highlights a clear distinction from power flow tasks, where edge features like line limits are crucial for modeling potential cascades and therefore have a significant impact on performance"

---

> > ### Comment · Reviewer_qprq · 2024-08-20
> > **Since cascading failures can have wide-reaching effects, I'm curious why long-range dependencies aren't considered relevant in this dataset**
> >
> > I want to thank the authors for their reply. Where do I find the GBT results on the webpage https://powergraph.ivia.ch?
> >
> >
> > > The reason might be due to the size of our grid and the well-known oversmoothing phenomena in GNN. In the cited reference, they tested GNN on distribution networks, while we focus on Transmission networks, entailing an entirely different topology and properties.
> >
> > I guess that statement refers to the number of GNN layers.  Do I understand the authors correctly, that they assume that long range dependencies are only relevant at distribution grid level, but not at transmission level? I can not agree with that assumption. Cascading failures are not necessarily local events. There are numerous examples of cascades across long distances. For example, the outage in Europe (https://en.wikipedia.org/wiki/2006_European_blackout). There were consequences in countries such as Spain, Morocco and Greece after disconnecting a line in Northern Germany. We also know, from recent studies of the Texan power grid, that long range dependencies occur (https://www.nature.com/articles/s41560-023-01434-1).
> >
> >
> > Hence, we know that in reality, long-range dependencies are relevant. Apparently, they do not appear to be relevant in the new dataset. What are the causes? That makes me question, how relevant the datasets are. Do the authors believe that their power grid modeling does not adequately capture the relevant phenomena?
> >
> > Considering that GBT even outperform some GNN models at some tasks and considering the concerns by Reviewer EEeb: "The current methods exhibit relatively high test accuracy on this dataset, suggesting that the dataset may not present sufficient complexity for benchmarking more advanced models", I am wondering how useful the datasets are for benchmarking GNN models. At least for the node-level tasks when GBT outperforms GNNs, the node attributes are more relevant than topological patterns.

---

> > > ### Comment · Reviewer_qprq · 2024-08-20
> > > **Literature on cascading failures**
> > >
> > > Hi,
> > >
> > > maybe the authors are also interested in "Small vulnerable sets determine large network cascades in power grids" (https://www.science.org/doi/full/10.1126/science.aan3184).

---

> > > ### Author Rebuttal · Authors · 2024-08-20
> > >
> > > Thank you for your comments and for raising these important points. We appreciate the opportunity to clarify our findings and address your concerns.
> > >
> > > 1. Relevance of long-range dependencies in our dataset
> > >
> > > Firstly, we want to clarify that we did not intend to imply that long-range dependencies are irrelevant or absent in our dataset. We fully acknowledge the importance of long-range dependencies, especially in cascading failure events, as demonstrated in historical blackouts such as the 2006 European blackout and recent studies on the Texan power grid. Our statement about the effectiveness of a three-layer GNN refers specifically to our observation that a three-layer GNN architecture can perform well in the tasks we benchmarked, due to the nature of how GNNs aggregate information from neighbors. According to research (see A Survey on Graph Neural Network Acceleration: Algorithms, Systems, and Customized Hardware, https://arxiv.org/pdf/2306.14052), a three-layer GNN effectively aggregates information from all direct neighbors within two hops, which is sufficient for many of the node-level tasks we studied.
> > >
> > > However, we agree that cascading failures are complex phenomena that can involve both local disturbances and long-range effects, albeit the latter are generally less frequent (see Power blackouts in Europe: Analyses, key insights, and recommendations from empirical evidence in Joule, https://www.cell.com/joule/pdf/S2542-4351(23)00366-5.pdf). The author observe from real operational data from the Italian transmission grid, that most cascading failures are local events. Therefore, while the current GNN models we tested show promising results, particularly with three layers, we do not dismiss the potential for more complex models that could better capture these rare but critical long-range dependencies in transmission networks.
> > >
> > > 2. The role of GBT in node-level tasks
> > >
> > > Regarding the performance of Gradient Boosted Trees (GBT) on node-level tasks, we agree that the strong results achieved by GBT indicate that current GNN architectures may not yet fully leverage the topological features of the dataset. This highlights a key area for future research: developing more expressive GNN architectures or improving message-passing layers to better capture the underlying physics and topology of power grids. While GBT models excel in utilizing node attributes, they lack the ability to directly model the complex relationships inherent in the graph structure, which is where advanced GNNs could ultimately outperform them. The fact that simpler models like GBT perform well on node-level tasks suggests that the current GNN models need to be refined and enhanced, rather than indicating any shortcomings in the dataset. This further underlines the need for ongoing research to improve GNN architectures for power systems analysis.
> > >
> > > In summary, while our results show that simple GNN models can be outperformed by GBT in node-level tasks, this serves as a call to action for the community to enhance GNN architectures. As we work to refine PowerGraph, we are also committed to making all results accessible on our website. In the meantime, we have attached the relevant GBT results for your review. The results will be visible on our website shortly.

---

> > > > ### Comment · Reviewer_qprq · 2024-08-23
> > > > **Reply to authors - increasing my score**
> > > >
> > > > I want to thank the authors for their quick response! I agree that cascading failures can be local and since this benchmark dataset is the first of its kind, it serves as a good starting point. Given the importance of power grids, I would like to see a growing ML community working in this field. I consider this benchmark dataset as an important step in this direction. Hence, I am willing to increase my score. However, I still see some limitations of the dataset, for example I am not sure how helpful they are to benchmark new GNN models, because performances are already relatively good, and node features might be more important than topological patterns. Furthermore, the current small grid sizes might limit relevance for real-world applications. Hence, I am already looking forward to seeing further extensions of the datasets in the future, for example to see the results of the synthetic Texan grid.

---

> > > > > ### Author Response · Authors · 2024-08-23
> > > > > **Reply to reviewer qprq**
> > > > >
> > > > > Thank you for your thoughtful feedback and for considering an increase in your score! We agree that while this benchmark dataset is a crucial starting point for the ML community in the power grid domain, there are areas for improvement. We are actively working on the dataset's limitations and look forward to extending the dataset, particularly with the inclusion of the synthetic Texan grid, to enhance its value for the community. We are currently running cascading failure simulations with the Texan grid. Although we're facing some challenges with the data—such as lacking demand profiles and OPF not converging—we aim to have the first data structures ready in the coming weeks.

---

### Official Review · Reviewer_dGzJ · 2024-07-24

**Rating:** 7
**Confidence:** 2
**Correctness:** The claims made in the submission see…
**Clarity:** Yes.

**Review:**

Overall the paper is very well written and despite the complexity of the three problems it is trying to address, it remains clear and to the point with relevant information in an appendix for people unfamiliar with electrical power grid applications. It could be even more detailed as some parts remain unclear to me after a thorough read however the main messages of the paper are indeed very clear. The benchmark of GNN methods is very thorough as well as the XAI methods proposed for cascading failure anlyses. I think this paper should be accepted as it fills a clear gap in the literature and meets all criterias of a great work.

**Strengths:**

See Review.

**Additional Feedback:**

None

**Documentation:**

The GitHub repository where the codes are stored is quite dense and well articulated with three different sub projects for node tasks, graph tasks and explainability. A clear ReadME with the information needed to get started with the data (downloadable at another link) and the benchmark models. The appendices also gives additional and useful information.

**Ethics:**

No ethical concerns.

**Limitations:**

I did not identify any potential negative societal impact of the authors' work.

**Opportunities For Improvement:**

It would be interesting to give more details on how the dataset was created and justifying the assumptions taken by the authors in the simulation. Section A.3 of the appendix gives us some details but it still remains unclear to me what data we are looking at (even after downloading the .mat files provided by the authors). It may be clear for incumbents but less so for people unfamiliar with the application.

Also, I believe the choice of GNN models for the benchmark is comprehensive. However, GNN are not the only machine learning models that can address PF, OPF and cascading failure anlyses and other methods could have been included in the benchmark. I acknowledge the fact that the main goal of the authors is to propose a benchmark tailored for GNN but it would have been good to also make some room for some statistical models and less elaborate deep learning models just to give a sense of where GNNs stand against them. The authors do mention a relevant reference that concludes in favour of GNN for these tasks but this paper should also be an opportunity to reassess this finding on more hollistic and transparent data.

**Relation To Prior Work:**

Yes the relation to prior work is clearly detailed particularly in the introduction.

**Summary And Contributions:**

The paper presents PowerGraph, a collection of datasets and benchmarks for Power Flow, Optimal Power Flow and Cascading failure analyses of power grids.A thorough outlook on state of the art explainable GNN methods is presented for cascading failure analysis and a comprehensive benchmark of GNN models applied to node and graph-level tasks is also proposed.

---

> ### Author Rebuttal · Authors · 2024-08-16
>
> •Thank you for your valuable comment. We understand that navigating the dataset may pose challenges, particularly for those less familiar with the application. Section 2, including Figure 1 and Figure 2, aim to provide a clearer overview of the dataset creation process. To further clarify this, we recommend also Section C1, which we updated to address the comment providing a detailed explanation of the content in each raw data file:
>
> “PowerGraph is the collection of the following GNN datasets: UK, IEEE24, IEEE39, and IEEE118 power grids. We use InMemoryDataset [60] class of Pytorch Geometric, which processes the raw data obtained from the Cascades [61] simulation.
> For each dataset UK, IEEE24, IEEE39, and IEEE118, we provide a folder containing the raw data organized in the following files for node-level tasks, i.e., power flow and optimal power flow analyses:
>
> -edge_attr.mat: edge feature matrix for the power flow problem (branch conductance $G_{ij}$, branch susceptance $B_{ij}$.)
>
> -edge_attr_opf.mat: edge feature matrix for the optimal power flow problem (branch conductance $G_{ij}$, branch susceptance $B_{ij}$.)
>
> -edge_index.mat: list of branches, represented as connection from node - to node
>
> -edge_index_opf.mat: list of branches, represented as connection from node - to node
>
> -X.mat: node feature matrix for the power flow problem (active power generation $P_{g}$ - active power demand $P_{d}$, reactive power generation $Q_{g}$ - reactive power demand $Q_{d}$, voltage magnitude $V$ , and voltage angle $\theta$, the number of loads $N_{loads}$, and number of generators $N_{gen}$).
>
> -Xopf.mat:node feature matrix for the optimal power flow problem (active power generation $P_{g}$ - active power demand $P_{d}$, reactive power generation $Q_{g}$ - reactive power demand $Q_{d}$, voltage magnitude $V$ , and voltage angle $\theta$, the number of loads $N_{loads}$, and number of generators $N_{gen}$).
>
> -Y_polar.mat: node output matrix for the power flow problem (active power generation $P_{g}$ - active power demand $P_{d}$, reactive power generation $Q_{g}$ - reactive power demand $Q_{d}$, voltage magnitude $V$ , and voltage angle $\theta$).
>
> -Y_polar_opf.mat: node output matrix for the optimal power flow problem (active power generation $P_{g}$ - active power demand $P_{d}$, reactive power generation $Q_{g}$ - reactive power demand $Q_{d}$, voltage magnitude $V$ , and voltage angle $\theta$).
>
> For graph-level tasks, i.e., cascading failure analysis:
>
> -blist.mat: list of branches, represented as connection from node - to node.
>
> -of\_bi.mat: binary classification labels ($DNS=0$ or  $DNS\neq0$)
>
>  -of\_reg.mat: regression labels ($DNS$)
>
> -of\_mc.mat: multi-class classification labels (See Table3)
>
> -Bf.mat: node feature matrix (Net active power at bus $P_{net}$, Net apparent power at bus $S_{net}$, Voltage magnitude $V$
>
> -Ef.mat: edge feature matrix (Active power flow $P_{i,j}$, Reactive power flow $Q_{i,j}$, Line reactance $X_{i,j}$, Line rating $lr_{i,j}$)
>
> -exp.mat: ground-truth explanation (See Appendix A.5)”
>
> Furthermore, we also updated the readme file in GitHub repository https://github.com/PowerGraph-Datasets for completeness.
>
> •	Thank you for your valuable suggestion. Although our dataset was specifically designed to test GNN, we also added Gradient Boosted Trees for comparison. We also believe that there should be some space for other models besides GNN. The results are included in the leaderboard on our website https://powergraph.ivia.ch. Furthermore, we have added the following comments in the Observation and Discussion paragraph of Section 3, respectively:
>
> “The website also includes Gradient Boosted Trees results as a baseline for comparison.”
>
> “We observe the limitations of GNNs, particularly in node regression tasks, where simpler models like Gradient Boosted Trees (GBT) outperform GNNs (see powergraph.ivia.ch). However, this trend does not extend to graph-level tasks, where GNNs consistently prove to be the superior method. This highlights the need for further development of GNN models tailored specifically for node regression tasks, critical in power systems analysis. At the same time, even minor improvements in classification tasks should not be underestimated, as they can significantly impact decision-making in critical infrastructure.”
>
> As the GBT results may not be visible on the website before the deadline, they are presented here below for transparency.

---

### Official Review · Reviewer_EEeb · 2024-07-25

**Rating:** 6
**Confidence:** 3
**Correctness:** Yes, the claims made in the paper are…
**Clarity:** Yes, the paper is well organized and …

**Review:**

See below.

**Strengths:**

This dataset introduces a novel and versatile resource with significant applications. It represents the first real-world dataset offering ground-truth explanations for graph-level tasks, making it invaluable for benchmarking the accuracy and faithfulness of explainability methods.

The accompanying documentation is thorough and detailed, ensuring that subsequent researchers can easily reproduce results and evaluate their methodologies using this dataset.

Additionally, this dataset stands out as the only publicly accessible real-world graph dataset specific to power grids. As such, it holds considerable potential for benefiting both researchers and practitioners within the power grid community.

**Additional Feedback:**

No

**Documentation:**

Yes, the authors provide adequate materials for reproducibility.

**Ethics:**

No.

**Limitations:**

Yes.

**Opportunities For Improvement:**

Considering the critical role of temporal information in power grids, constructing temporal graphs through the simulation of cascading processes would be highly valuable.

The current methods exhibit relatively high test accuracy on this dataset, suggesting that the dataset may not present sufficient complexity for benchmarking more advanced models. This limitation may stem from the data generation process, which should introduce more potential confounders and randomness while maintaining a basis in physical principles.

A more detailed analysis of the experimental results is warranted. For instance, it would be beneficial to explore why one model might significantly outperform another due to specific properties of the dataset that only the former model accounts for.

**Relation To Prior Work:**

Yes.

**Summary And Contributions:**

This paper introduces PowerGraph, a graph dataset specifically designed to focus on cascading failure events in power grids. It presents a publicly accessible graph dataset for power grids. The authors claim that it is also the first real-world graph dataset to include reliable ground-truth explanations. PowerGraph is derived using a physics-based Cascades model to simulate cascading failures, making it versatile for various applications: i) benchmarking Graph Neural Network (GNN) models for graph-level tasks such as multi-class classification, binary classification, and regression, and ii) benchmarking GNN explainability methods by providing ground-truth explanations.

---

> ### Author Rebuttal · Authors · 2024-08-16
>
> •The cascading stage information is already included in the explainability.mat file, allowing for the straightforward reconstruction of the temporal graph. Given the approaching deadline, it would not be feasible to benchmark temporal GNN methods across all datasets, as re-running the experiments would take at least several weeks. We have added a section in the appendix:
> "Constructing Temporal Graphs to Analyze the Cascading Process in PowerGraph." This section provides detailed instructions on how to utilize the current data to perform these experiments: “Each instance of the graph-level dataset in PowerGraph, designed for cascading failure analysis, is accompanied by an explainability mask that encodes the stages of the cascading failure process. This mask is stored in a MATLAB structure exp.mat (see Appendix C.1). Each entry in the MATLAB structure is a vector containing all failures at all stages, with the order of the vector corresponding to the sequence of line failures in the cascading event. Using this information, a temporal graph can be reconstructed from PowerGraph.”
>
> We are committed to incorporating temporal graph construction and benchmarking temporal graph neural networks in  PowerGraph. This enhancement will significantly contribute to a more comprehensive analysis in future research. We adapted the conclusion paragraph to mention the future work:
>
> “Looking forward, we plan to enhance PowerGraph by adding a temporal graph dataset to facilitate in-depth analysis of cascading failures. Nevertheless, the current dataset contains sufficient information to create such type of data. We will also benchmark methods on larger synthetic power systems, such as the grid in [49], to assess whether deeper GNN architectures are necessary, as noted in [50]. A raw dataset is available at https://powergraph.ivia.ch.”
>
> •	Thank you for your comment. It is important to emphasize that although a high performance in the classification tasks is a significant outcome of this study, it is the least complicated problem formulation. On the other hand, the regression models for cascading failures, power flow, and optimal power flow do not yet achieve the desired performance levels. This reflects the inherent complexity of these phenomena, which are more realistically represented through regression. Therefore, we highlight the pressing need for the geometric deep learning community to advance GNN architectures. Hence, it provides valuable contributions not only to the field of GNNs but also to power grid research as a whole. As detailed in Appendices 2 and 3, as well as Section 4, our data generation process was meticulously designed to incorporate realistic demand profiles for power flow and optimal power flow models alongside a comprehensive range of single and multiple failures for the cascading failure dataset. While introducing additional randomness might increase the dataset's complexity, it could also compromise the realism essential in the field of electrical engineering. This is further described in the Discussion paragraph of Section 3 and in the Conclusions, respectively:
>
> “While binary and multi-class classification models show good results, the regression model predicting
> the exact demand not served does not. The R2 score, also known as the coefficient of determination, reaches a maximum of 0.43 for the best regression model on the IEEE24 dataset but remains below 0.26 for other power systems. Ideally, an R2 score closer to 1 is desired for a better model fit. This highlights the need for future research to focus on improving GNN performance on regression tasks both at the graph and node-level. We observe the limitations of GNNs, particularly in node regression tasks, where simpler models like Gradient Boosted Trees (GBT) outperform GNNs (see powergraph.ivia.ch). However, this trend does not extend to graph-level tasks, where GNNs consistently prove to be the superior method. This highlights the need for further development of GNN models tailored specifically for node regression tasks, which are critical in power systems analysis. At the same time, even minor improvements in classification tasks should not be underestimated, as they can significantly impact decision-making in critical infrastructure.”
>
> “Through rigorous benchmarking against a range of GNN and explainability models, PowerGraph exhibits high performance in graph classification, indicating a need for further refinement in regression models. Regression models remain essential in power systems and can be solved by GNNs for tasks like power flow and system security analysis. However, GNN still encounters challenges that demand further research and development.”
>
> •	To address your suggestion, we have added a clarification to the Discussion paragraph of Section 3:
>
> “Notably, for node-level tasks, the GAT model generally outperforms others. Nevertheless, the GCN model, which disregards edge features, also shows strong results without significantly impacting performance. The best results are consistently achieved with a single MPL across all cases. For graph-level tasks, the GCN model consistently shows lower performance compared to other models. In contrast, the Transformer and GINe models emerge as the best performers. This highlights a clear distinction from power flow tasks, where edge features like line limits are crucial for modeling potential cascades and therefore have a significant impact on performance. The Transformer model leverages its attention mechanism to dynamically weigh the significance of neighboring nodes, capturing complex relationships and leading to superior learning outcomes. This attention-based approach is a key factor in the Transformer’s and GAT’s effectiveness across both node-level and graph-level tasks. Additionally, the GINe model demonstrates robust performance due to its strong capability in capturing and aggregating local neighborhood information, which enhances its predictive accuracy [29].”

---

> > ### Comment · Reviewer_EEeb · 2024-08-28
> >
> > Thanks for your rebuttal. I would like to keep my score as positive.

---

> > > ### Author Response · Authors · 2024-08-30
> > > **Reply to Official Comment by Reviewer EEeb**
> > >
> > > We sincerely thank the reviewer for their valuable feedback. By incorporating their suggestions—particularly the addition of detailed explanations on the creation of temporal graphs with our dataset and a more thorough discussion of our results—we have meaningfully enhanced the overall quality and completeness of our paper.

---

### Decision · Program_Chairs · 2024-09-26

**Decision:**

Accept (Poster)

**Comment:**

The four reviewers are supportive to this submission, the first of its kind, as can be viewed from the comments and posts.  There are some issues suggested for further exploration, for example, larger grids, long-range dependencies in the datasets, temporal graphs. Hope the final version will reflect the reviewers comments, and more can be included in the website  https://powergraph.ivia.ch.